

# Entanglement entropy from entanglement contour: Higher dimensions

**Muxin Han[1,2] and Qiang Wen[3,4⋆]**

**1** Department of Physics, Florida Atlantic University, 777 Glades Road,
Boca Raton, FL 33431-0991, USA
**2** Institut für Quantengravitation, Universität Erlangen-Nürnberg,
Staudtstr. 7/B2, 91058 Erlangen, Germany
**3** Shing-Tung Yau Center of Southeast University, Nanjing 210096, China
**4** School of Mathematics, Southeast University, Nanjing 211189, China

⋆ wenqiang@seu.edu.cn

## Abstract

We study the *entanglement contour* and *partial entanglement entropy* (PEE) in quantum field theories in 3 and higher dimensions. The entanglement entropy is evaluated from a certain limit of the PEE with a geometric regulator. In the context of the *entanglement contour*, we classify the geometric regulators, study their difference from the UV regulators. Furthermore, for spherical regions in conformal field theories (CFTs) we find the exact relation between the UV and geometric cutoff, which clarifies some subtle points in the previous literature. We clarify a subtle point of the additive linear combination (ALC) proposal for PEE in higher dimensions. The subset entanglement entropies in the *ALC proposal* should all be evaluated as a limit of the PEE while excluding a fixed class of local-short-distance correlation. Unlike the 2-dimensional configurations, naively plugging the entanglement entropy calculated with a UV cutoff will spoil the validity of the *ALC proposal*. We derive the *entanglement contour* function for spherical regions, annuli and spherical shells in the vacuum state of general-dimensional CFTs on a hyperplane.

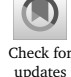

# 1 Regulators for entanglement entropy in quantum field theories

The ideas of quantum information theory have intersected more and more deeply with our understanding of quantum field theory, string theory and quantum gravity. One of the most important quantities is the entanglement entropy, which captures the bipartite entanglement in a pure state $|\Psi\rangle$. Dividing a quantum system into $A \cup \bar{A}$, the entanglement entropy between $A$ and $\bar{A}$ can be calculated by the von Newmann entropy of the reduced density matrix $\rho_A = \mathrm{Tr}_{\bar{A}} |\Psi\rangle\langle\Psi|$,

$$S_A = -\mathrm{Tr}\rho_A \log \rho_A. \tag{1}$$

In condensed matter physics, the entanglement entropy has been used to distinguish new topological phases and characterize critical points, e.g., [1–5]. In high energy physics the Ryu-Takayanagi (RT) formula [6, 7] relates quantum entanglement to spacetime geometry in the context of AdS/CFT [8–10] (the most well studied holography), which states that the $(d+1)$-dimensional gravity theories with asymptotically AdS boundary is equivalent to certain $d$-dimensional conformal field theories on the boundary. Hence the entanglement entropy becomes an important tool to study quantum gravity via holography, and furthermore the holography itself. Many important progresses along this line are cited and summarized in the following review articles [11–16] and the recently published book [17].

Unlike its definition, the calculation of the entanglement entropy is a formidable task. Furthermore, the entanglement entropy in quantum field theory is divergent because of the collection of the endless short-distance correlations, hence cannot be well-defined without a regulator. The role of the regulator is to subtract the divergent short-distance correlations from the entanglement entropy. However, the way we perform the regularization is highly non-unique. This will result in different values for the regularized entanglement entropy under different regularization schemes. The discussion on different types of regulators in the previous literature is far from clear. The mixing between the regulators has already caused ambiguities in our understanding of the entanglement entropy in quantum field theories. In this paper, we will classify two different types of regulators, study their differences and show how they exactly match with each other in the special cases where the matching is possible.

The first type of regulator is to regulate the entanglement entropy at a small scale $\delta$. Roughly speaking this regulator excludes all short-distance correlations below the scale $\delta$ when we "count" the entanglement between the region $A$ and its complement $\bar{A}$. This regulator is

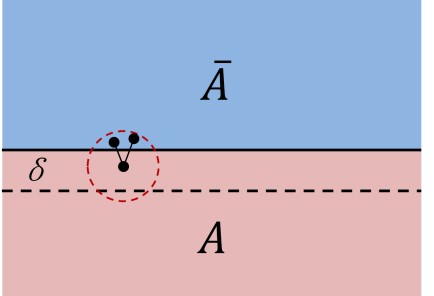 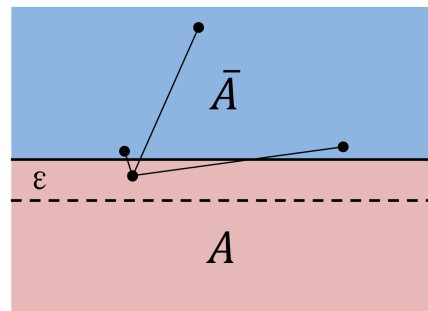

Figure 1: The $\delta$ neighborhood of a point $P$ consists of all the points whose distance from $P$ is smaller than $\delta$ (see the region inside the dashed red circle). In the left figure, the UV regulator excludes the correlation between $P$ and those inside the overlap between its $\delta$ neighborhood and $\bar{A}$. While in the right figure, the geometric regulator exclude the correlation between $P$ (in certain cutoff region) and all the degrees of freedom inside $\bar{A}$

depicted in the left figure of Fig. 1. For any point whose distance from the boundary of $A$ is smaller than $\delta$, we exclude the correlation between degrees of freedom at this point and those inside a subregion of $\bar{A}$, which is the overlap between its $\delta$ neighborhood (region inside the red dashed circle) and $\bar{A}$. This type of regulator is familiar to us and the cutoff $\delta$ is usually called the UV cutoff. They have been applied in the replica trick [3, 4, 18] in quantum field theories and the Ryu-Takayangi (RT) formula [6, 7, 19] for holographic CFTs, which are among the most important ways to evaluate the entanglement entropy. The RT formula provides a simple and elegant way to calculate entanglement entropies in the context of AdS/CFT. More explicitly, for a subregion $A$ in the boundary CFT and a minimal surface (or more generally an extremal surface) $\mathcal{E}_A$ in the dual AdS bulk spacetime, where $\mathcal{E}_A$ is anchored on the boundary $\partial A$ of $A$, the RT formula states that the entanglement entropy of $A$ is measured by the area of $\mathcal{E}_A$ in Planck units

$$S_A = \frac{Area(\mathcal{E}_A)}{4G}, \tag{2}$$

which provides a simpler routine to calculate entanglement entropy. In AdS/CFT the radius coordinate $z$ represents the scale along the RG flow and the boundary is settled at $z = 0$. The minimal surface $\mathcal{E}_A$ is cut off at $z = \delta$ hence the entanglement entropy, which is related to its area, is regulated at the scale $\delta$.

The second type of regulator is the geometric regulator. This type of regulator is based on quantities that capture certain types of correlation between two spacelike separated regions $A$ and $B$. Usually the two regions do not share boundaries and $A \cup B$ is in a mixed state. It can be purified by another system $C$, hence $A \cup B \cup C$ is in a pure state (see the left figure of Fig. 2). Let us write this quantity as $\mathcal{C}(A, B)$. To be a good candidate to regulate the entanglement entropy, firstly $\mathcal{C}(A, B)$ should be well defined hence is free from divergences, and secondly $\mathcal{C}(A, B)$ should possess the property of normalization,

$$\mathcal{C}(A, B)|_{B \to \bar{A}} \to S_A. \tag{3}$$

In the above prescription we let $B$ approach $\bar{A}$ while keeping $A$ fixed. A more general configuration is to let both sides of the entangling surface approach. Consider an entangling surface that divides the total system into $A \cup \bar{A}$. Then we consider two regulated regions $A_{reg} \subset A$ and $\bar{A}^{reg} \subset \bar{A}$ and $C$ being an infinitely narrow strip that contains the entangling surface between $A$ and $\bar{A}$ (see the right figure of Fig. 2). The normalization is then given by the following

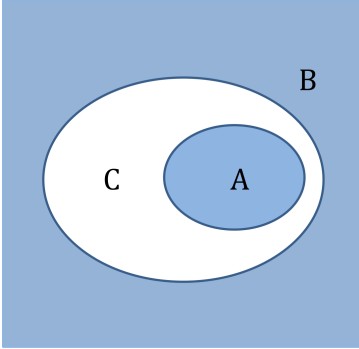 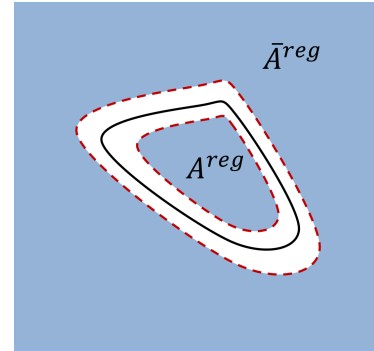

Figure 2: The quantity $\mathcal{C}(A, B)$ approaches the entanglement entropy under the geometric regulator. In the right figure, the black solid line is the boundary between the region $A$ and its complement $\bar{A}$. We regularize these two regions to be $A^{reg}$ and $\bar{A}^{reg}$, while the white region is the cut-off region $A^{cut} \cup \bar{A}^{cut}$ which should contain the boundary. The red dashed lines are the boundaries between regularized regions and the cutoff regions.

requirement,

$$\mathcal{C}(A^{reg}, \bar{A}^{reg})|_{A^{reg} \to A,\ \bar{A}^{reg} \to \bar{A}} \to S_A. \tag{4}$$

The width of the strip $C$ should be lower bounded by a small constant $\epsilon$ to avoid divergence in $\mathcal{C}(A^{reg}, \bar{A}^{reg})$. Note that, this type of regulator is very sensitive to how $A^{reg}$ ($\bar{A}^{reg}$) approaches $A$ ($\bar{A}$) as well as the geometric information of the entangling surface. Usually we will set the width of the region $C$, as well as the distance between the entangling surface and the boundaries of $C$, to be a constant.

The geometric regulators (4) are definitely different from the UV cutoff which exclude short distance correlation below a certain scale everywhere. The geometric regulators not only exclude the short-distance correlations within $C$ but also the long-distance correlations between the degrees of freedom in $C$ and those inside $A \cup B$ (see the right figure in Fig. 1). In other words, one may regard the UV cutoff to be a cutoff in the energy-momentum space while the geometric cutoff to be a cutoff in spacetime.

So far, there are two different quantities that have been proposed to be a natural regulator for entanglement entropy under the geometric regulator (4),

- the mutual information (MI) $\frac{1}{2}I(A, B)$[1] [22];

- the reflected entropy $\frac{1}{2}S_R(A, B)$ [23];

The treatment in [22] using MI is quite careful and successful for disk regions in 2+1 dimensional theories. They fix the width of $C$ to be an infinitesimal constant $\epsilon$, then they find that $I(A^{reg}, \bar{A}^{reg})$ has a similar expansion as the entanglement entropy, with an area term at the order $\mathcal{O}(\epsilon^{-1})$ and an universal term $c_0$. By carefully putting the entangling surface at the middle of the strip $C$, they demonstrated that the contribution from the high energy physics to the MI $I(A^{reg}, \bar{A}^{reg})$ will not pollute the universal term $c_0$ in entanglement entropy. Hence they find a well-defined way to determine the universal term $c_0$ in the entanglement entropy of a disk with a UV cutoff.

Also there is a similar prescription using the reflected entropy in [23]. The entanglement wedge is the region bounded by $A$, $B$ and the RT surface $\mathcal{E}_{A \cup B}$. Let the mixed state $\rho_{AB}$ be

---

[1]Although the mutual information could be considered as a regulator for the entanglement entropy, it contains information that are not encoded in the entanglement entropy [20, 21].

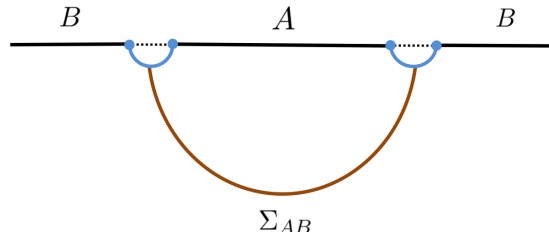

Figure 3: The boundary field theory is in a pure state and the complement of $A \cup B$ is infinitesimal. The blue lines consist of the RT surface $\mathcal{E}_{A \cup B}$ and the brown line $\Sigma_{AB}$ is the minimal cross section of the entanglement wedge $\mathcal{W}_{AB}$.

purified by an identical system $A^* B^*$ with the same copy of Hilbert space and partition, the reflected entropy is then defined as the entanglement entropy $S_R(A, B) = S_{AA^*}$ (see [23] for details). It was proved that half of the reflected entropy $S_R(A, B)$ is given by the area of the minimal cross section of the entanglement wedge (EWCS) $\Sigma_{AB}$,

$$\frac{1}{2} S_R(A, B) = \frac{Area(\Sigma_{AB})}{4G} \,. \tag{5}$$

The way how the reflected entropy acts as a natural regulator for entanglement entropy $S_A$ is shown in Fig. 3. Let $C$ be the complement of $A \cup B$ and the whole system on the AdS boundary $A \cup B \cup C$ is in a pure state. The RT surface $\mathcal{E}_{A \cup B}$ is given by the two blue minimal surfaces, while $\Sigma_{AB}$ is given by the solid brown minimal surface vertically anchored on $\mathcal{E}_{A \cup B}$. When we take the distance between $A$ and $B$ to be infinitesimal, the EWCS approaches the RT surface $\mathcal{E}_A$ and is terminated by $\mathcal{E}_{A \cup B}$ at a small $z = \delta$.

Based on the recent studies on the so-called *entanglement contour* [24] or *partial entanglement entropy* (PEE) [25–27], in this paper we propose the PEE to be the third quantity that can be taken as a natural regulator for the entanglement entropy under a geometric regulator. The *entanglement contour* characterizes the contribution from any site inside $A$ to the entanglement entropy $S_A$. The additivity and normalization make the PEE a natural regulator for the entanglement entropy. The geometric regulator excludes the contribution from those degrees of freedom near the boundary, which is the origin of the divergence. For example in the right figure of Fig. 1 we exclude the contribution from all the points whose distance from the boundary is smaller than $\epsilon$. The evaluation of the entanglement entropy from PEE in $d \geq 3$ is the main topic of this paper.

The main structure of the paper is in the following. In section 2, we will briefly introduce the concept of *entanglement contour* and PEE and review recent progress along this direction. Furthermore, we will discuss the relation between the PEE and the so-called extensive mutual information (EMI). In section 3, we discuss how to use the *entanglement contour* to evaluate the entanglement entropy by taking a geometric cutoff. Unlike the previous studies, here we focus on the *entanglement contour* in higher dimensions ($d \geq 3$), where a subtlety to apply the *ALC proposal* for PEE needs to be clarified. In section 4 we derive the *entanglement contour* function for static spherical regions in vacuum CFT in arbitrary dimensions. We firstly derive it via the Rindler method [28], then we re-derive the contour function via the fine structure analysis of the entanglement wedge generalized from [25]. In section 5, based on the *ALC proposal* (12) and the *entanglement contour* function for spherical regions, we propose a prescription to derive the *entanglement contour* for annuli and spherical shells. In section 6, we

consider the special case of spherical region and give an exact matching between the UV cutoff and geometric cutoff, hence exactly matching the entanglement entropy evaluated by the RT formula and the one evaluated by the Rindler method. In section 7, we summarize the main points of this paper.

## 2 The entanglement contour and partial entanglement entropy

The *entanglement contour* firstly introduced in [24] is a function that captures how much the degrees of freedom at each site of $A$ contribute to the total entanglement entropy $S_A$. It gives a local measurement of the spatial structure of quantum entanglement. We denote this function as $s_A(\mathbf{x})$, where $\mathbf{x}$ denotes the points in the region $A$. By definition in $d$-dimensional theories it satisfies the following basic requirement

$$S_A = \int_A s_A(\mathbf{x})dx^{(d-1)}. \tag{6}$$

Instead of studying the contour function directly, it is more convenient to study the *partial entanglement entropy* (PEE) $s_A(A_i)$ for any subset $A_i$ of $A$, which captures the contribution from $A_i$ to $S_A$ and is defined by

$$s_A(A_i) = \int_{A_i} s_A(\mathbf{x})dx^{(d-1)}. \tag{7}$$

In this paper we evaluate the entanglement entropies as a limit of the PEE. Also based on the *additive linear combination proposal* [25, 26] and the *entanglement contour* for some highly symmetric regions, we can evaluate the entanglement entropy or *entanglement contour* for certain subregions with less symmetries, like annuli and spherical shells.

However, the fundamental definition based on the reduced density matrix for the PEE is still missing. According to its assumed physical meaning, the PEE $s_A(A_i)$ should capture certain type of the correlation between the subregion $A_i$ of $A$ and any complement system $\bar{A}$ that purifies $A$. In order to manifest its role as certain correlation between two regions, it is more convenient to write is in the following way

$$s_A(A_i) = \mathcal{I}(A_i, \bar{A}). \tag{8}$$

Be careful not to confuse the PEE $\mathcal{I}$ with the MI $I$. The expression $s_A(A_i)$ is more convenient to show the contribution distribution for $S_A$, while $\mathcal{I}(A_i, \bar{A})$ is more convenient to show the correlation structure between spacelike separated regions.

The physical meaning of the PEE indicates that it should satisfy certain physical requirements[2], which we write in terms of $\mathcal{I}(A, B)$ in the following:

1. *Additivity*: if $B \cap C = \emptyset$ and both of $B$ and $C$ are spacelike separated from $A$, by definition we should have

$$\mathcal{I}(A, B \cup C) = \mathcal{I}(A, B) + \mathcal{I}(A, C). \tag{9}$$

2. *Invariance under local unitary transformations*: $\mathcal{I}(A, B)$ should be invariant under any local unitary transformations inside $A$ or $B$.

3. *Symmetry*: for any symmetry transformation $\mathcal{T}$ under which $\mathcal{T}A = A'$ and $\mathcal{T}B = B'$, we have $\mathcal{I}(A, B) = \mathcal{I}(A', B')$.

---

[2]The requirements 1-6 are firstly given in [24], while the requirement 7 is recently given in [27].

4. *Normalization*: $\mathcal{I}(A,B)|_{B\to\bar{A}} \to S_A$.

5. *Positivity*: $\mathcal{I}(A,B) \geq 0$.

6. *Upper bound*: $\mathcal{I}(A,B) \leq \min\{S_A, S_B\}$.

7. *The permutation symmetry between A and B*: $\mathcal{I}(A,B) = \mathcal{I}(B,A)$.

The requirement of the upper bound can be derived from the requirements 1, 4 and 5 hence is not independent. The requirement 7 is based on the assumption that $\mathcal{I}(A,B)$ is a measure of the correlation between $A$ and $B$. This requirement gives non-trivial constraints for the PEE. For example consider the configuration in the left panel of Fig. 2, the requirement 7 implies,

$$s_{A\cup C}(A) = s_{B\cup C}(B). \tag{10}$$

The additivity significantly simplified the entanglement structure of the system. All PEEs are just a double summation of the elementary PEEs between two arbitrary sites,

$$\mathcal{I}(A,B) = \sum_{i\in A,\ j\in B} \mathcal{I}(i,j), \tag{11}$$

where $\mathcal{I}(i,j)$ is the PEE between the $i$th and $j$th site. In continuous systems like QFTs, the summation is replaced by integration and the PEE $\mathcal{I}(i,j)$ is replaced by the function $\mathcal{I}(\mathbf{x},\mathbf{y})$, where $\mathbf{x}$ and $\mathbf{y}$ are points in $A$ and $B$ respectively. In this paper we will focus on the *entanglement contour* or PEE in $d$-dimensional theories with $d \geq 2$.

The Gaussian formula [24, 29–35] applies to the Gaussian states in free theories. This is the first attempt to construct the *entanglement contour*. In the context of holography, a geometric construction [25, 36] gives an one-to-one correspondence between points on the boundary region $A$ and the points on the RT surface $\mathcal{E}_A$, hence gives the *entanglement contour* function for $A$. We will introduce this holographic picture later in section 4. In the following, we will briefly introduce the ALC proposal and the EMI formula for the PEE. So far the PEE calculated by all the existing proposals are highly consistent with one another, see for example [25, 27, 35, 36]. It implies that the PEE should be unique and well-defined.

**The additive linear combination proposal for PEE**

In [25, 26], a simple proposal for the *partial entanglement entropy* is given, which claims that the PEE is given by a linear combination of entanglement entropies of certain subsets in $A$. It was shown [25–27, 37] to satisfy all the above 7 requirements using only the general properties of entanglement entropy. It can be applied to generic theories, but a definite order is required for the degrees of freedom in $A$ in order to satisfy the additivity.

- *The additive linear combination proposal (ALC)*: Given a region $A$ and an arbitrary subset $\alpha$, when there is a definite order inside $A$, thus it can be partitioned into three non-overlapping subregions $A = \alpha_L \cup \alpha \cup \alpha_R$ (one can consider the typical example where $A$ is an interval divided into three subintervals) unambiguously, where $\alpha_L$ ($\alpha_R$) is denoted as the subset on the left (right) hand side of $\alpha$. In this configuration, the *ALC proposal* claims that

$$s_A(\alpha) = \frac{1}{2}\left(S_{\alpha_L\cup\alpha} + S_{\alpha\cup\alpha_R} - S_{\alpha_L} - S_{\alpha_R}\right). \tag{12}$$

The *ALC proposal* can be used to calculate the *entanglement contour* for one dimensional regions in a generic theory, since there is a natural order along the spatial direction [25, 26].

In higher dimensions, the natural order disappears. One may give an order to all the degrees of freedom inside *A* by hand. However, the PEE calculated by the *ALC proposal* depends on the order we give. Since the order is highly non-unique, the PEE calculated by the *ALC proposal* becomes ambiguous. In higher dimensions the *ALC proposal* only applies for highly symmetric configurations, where the contour function only depends on one coordinate as the order can be naturally defined along that coordinate. See Fig. 5 for an example in 3-dimensions where the *entanglement contour* only depends on the radial coordinate.

### The extensive mutual information formula for PEE

Note that, the requirements 2-7 are also satisfied by half of the MI $\frac{1}{2}I(A,B)$. Furthermore the additivity and positivity indicates that the PEE satisfies other two properties satisfied by the MI, the monotonicity

$$\mathcal{I}(A,BC) = \mathcal{I}(A,B) + \mathcal{I}(A,C) \geq \mathcal{I}(A,B), \tag{13}$$

and the Markov property (saturated)

$$\mathcal{I}(A,BC) + \mathcal{I}(A,CD) = \mathcal{I}(A,C) + \mathcal{I}(A,BCD). \tag{14}$$

In summary the physical requirements for the PEE coincide with the known ones for the MI plus the additivity.

In [38] the authors showed that, in Pincaré invariant theories the 7 requirements have a unique solution. If we further impose the conformal invariance this solution is explicitly given by the following formula

$$\mathcal{I}(A,B) = \kappa_{(d)} \int_{\partial A} d\sigma_A \int_{\partial B} d\sigma_B \frac{(n_A \cdot n_B)(\bar{n}_A \cdot \bar{n}_B) - (n_A \cdot \bar{n}_B)(\bar{n}_A \cdot n_B)}{|x_A - x_B|^{2(d-2)}}, \tag{15}$$

where $\partial A$ and $\partial B$ are the co-dimensional two boundaries of *A* and *B*, $\sigma_A$ ($\sigma_B$) is the area element on $\partial A$ ($\partial B$), $n_A$ and $\bar{n}_A$ are unit vectors orthogonal to the surface $\partial A$ and to each other $n_A \cdot \bar{n}_A = 0$. The constant $\kappa_d$ could be determined by the requirement of normalization. The formula can also be arrived at by assuming the twist operators to be exponentials of free fields on the boundaries $\partial A$ and $\partial B$ [39].

This formula (15) was originally used to describe the EMI, which is assumed to be the MI that is additive and may exist in some unknown theories. However, in [40] it was shown that even though the EMI satisfies all known requirements of MI in QFT, it does not correspond to the MI of any actual QFT or limit of QFT[3]. This then implies that the known properties satisfied by MI are not enough to confine their solution to be the MI of actual QFTs. In [27] the EMI formula was also written as a linear combination of the subset entanglement entropies. This linear combination coincides with the *ALC proposal* rather than the MI. Hence from our point of view, in general CFTs it is more natural to interpret the EMI formula (15) as the PEE rather than the MI.

The EMI formula for the PEE has many advantages compared with the *ALC proposal*. Firstly, it does not rely on the natural order for all the degrees of freedom inside *A*. Secondly, it applies to a connected region in general dimensions with arbitrary shape. The disadvantage is that it only applies to CFTs while the *ALC proposal* applies to general theories. The calculations of $\mathcal{I}(A,\bar{A})$ for CFTs in various configurations have been carried out [41–44] using the EMI

---

[3]Except the free fermions in two dimensions [38].

formula, and the results reproduce (not exactly) the expected behavior for the entanglement entropies compared with those evaluated with a UV cutoff.

The bit threads configuration [45] also gives a natural picture for the *entanglement contour* in holographic theories. For a given region $A$ and a bit thread configuration, the PEE $s_A(A_i)$ can be read by counting how many bit threads emanating from $A_i$ eventually cross the RT surface $\mathcal{E}_A$ and anchor on $\bar{A}$. Nevertheless, the bit threads configurations that give the maximal flux through $A$ are highly non-unique [45]. This conflicts with our expectation that the *entanglement contour* should be unique. We suggest that further requirements should be imposed to bit threads in order to give the right *entanglement contour*. This is recently explored in [46] by applying the locking theorem [47,48] of bit threads to construct a concrete locking scheme for the RT surfaces in the entanglement wedge. Other exploration of *entanglement contour* based on a specific bit thread configuration can be found in [37,49,50].

The *entanglement contour* gives a finer local measure for the entanglement structure. It has been shown to be particularly useful to characterize the spreading of entanglement when studying dynamical situations [24,34,37,51]. The modular flow in two dimensions can be generated from the PEE [26]. The *entanglement contour* is also a useful probe of slowly scrambling and non-thermalizing dynamics for some interacting many-body systems [52]. Holographically, the PEE [25,36] corresponds to bulk geodesic chords which is a finer correspondence between quantum entanglement and bulk geometry [36,53]. Under some balanced condition, the PEE also gives the area of the entanglement wedge cross section [54]. The balanced PEE can be considered to be a generalization of the reflected entropy [23] to generic purifications of the bipartite system [54]. The first law of the *entanglement contour*, which captures local perturbation of the entanglement structure, was studied in [55]. In [51,56], the *entanglement contour* captures the spatially fine-grained picture of the entanglement structure of Hawking radiation, and is shown to be vanished for certain regions when the non-trivial island appears, hence gives more information than the page curve.

## 3 Entanglement entropy from entanglement contour and the ALC proposal in higher dimensions

### 3.1 PEE as a natural regulator of entanglement entropy

Since the distance between the degrees of freedom in $A^{reg}$ and those in $\bar{A}^{reg}$ are bounded from below, the PEE $\mathcal{I}(A^{reg}, \bar{A}^{reg})$ should be free from divergence. In the special cases where $A = \alpha_L \cup \alpha \cup \alpha_R$ with the subsets in a natural order (for example when $A$ is a segment), $\mathcal{I}(\bar{A}, \alpha)$ can be given by the *ALC proposal* (12). Like the MI, the divergent area terms cancel with each other. Also, if we let $\alpha_L$ and $\alpha_R$ approach the empty space, the PEE will recover the entanglement entropy $S_A$. Furthermore, it was recently pointed out in [51] that the linear combination in the *ALC proposal* is just a conditional mutual information

$$\mathcal{I}(\bar{A}, \alpha) = \frac{1}{2} I(\alpha, \bar{A}|\alpha_L) = \frac{1}{2} I(\alpha, \bar{A}|\alpha_R), \tag{16}$$

which is a well-defined quantity in quantum information.

Because of the additivity and normalization, it is quite natural using the PEE to evaluate the entanglement entropy under the geometric regulator (4). This is similar to the prescription of [22] using the MI, but the way the PEE approaches the EE should be different from the MI as they are different quantities. Since the PEE $s_A(A_i)$ is defined to capture the contribution from the subset $A_i$ to $S_A$, it is straightforward to claim that the entanglement entropy $S_A$ can be recovered by collecting contributions from all the degrees of freedom inside $A$, i.e. taking

the limit $A_i \to A$ for $s_A(A_i)$. Note again that the geometric regulator not only excludes the short distance correlation across the boundary, but also the long range correlations between the cutoff region and $A^{reg} \cup \bar{A}^{reg}$.

In the previous literature, the entanglement entropies evaluated from the geometric regulator are usually used to reproduce the results with a UV cutoff. Our discussion on the difference between the UV regulator and the geometric regulator implies that this matching does not have a clear physical motivation. Subtleties or ambiguities will appear when we try to do the matching. For example, the ambiguity to reproduce the entanglement entropy evaluated from the RT formula using the Rindler method already appears in [28, 57]. In holographic CFTs the Rindler transformations map the static spherical region $0 < r < R$ to the Rindler space $0 < u < \infty$ with infinitely far away boundaries (see section 4 for details). Since the Rindler transformations are symmetries of the theories, the entanglement entropy for the spherical region equals to the thermal entropy of the Rindler space. Due to the translation symmetries in the Rindler space, the entropy density (which is a contour function) is a constant that can be evaluated holographically. The thermal entropy is then regulated by introducing a cutoff for the volume of the Rindler spacetime. This is indeed a geometric regulator for the entropy via the PEE. More explicitly we exclude the contribution from the region $u > u_{max}$. The region in the original spacetime that is mapped to $u > u_{max}$ is just $R - \epsilon < r < R$, where $\epsilon$ is determined by $u_{max}$ and is infinitesimal when $u_{max}$ is infinitely large. If one take this $\epsilon$ to be the UV cutoff $\delta$, the results from the Rindler method will not be consistent with those from the RT formula with the RT surface cut off at $z = \delta$.

Also, in the appendix D.1 of [22], the authors considered a free massless scalar with $d = 3$ and studied the entanglement entropy for a disk region via the Rindler method. In this case, the entropy density in the Rindler spacetime can be obtained via the heat kernel method. Again one can evaluate the entanglement entropy by regulating the volume of the Rindler spacetime with $u_{max}$. When mapping back to the original spacetime, this prescription is just evaluating the entanglement entropy from the PEE with a geometric regulator $\epsilon$. Expanding the entanglement entropy with respect to $\epsilon$, we will find that the term $c_0$ at the order $\mathcal{O}(\epsilon^0)$ is not the universal term $c_0^{scalar}$ in the entanglement entropy with a UV cutoff [57, 58]. Instead, they are related by the following relation,

$$c_0 = \frac{3}{2} c_0^{scalar}. \tag{17}$$

In [28, 57], this ambiguity was hidden by choosing another cutoff for the volume of the Rindler spacetime, which we call $u'_{max}$. It is determined by the requirement that the cutoff points with $u = u'_{max}$ on the horizon in the Rindler bulk spacetime are mapped to the points on the RT surface with $z = \delta$. This choice reproduces the entanglement entropy from the RT formula. However, it is not natural in the Rindler method as it contains further input from holography. On the field theory side the points with $u = u'_{max}$ are mapped to the points with $r = R - \epsilon$ with $\epsilon \neq \delta$. In other words, in order to reproduce the entanglement entropy with a UV cutoff $\delta$, we should collect the contribution from the degrees of freedom in $0 < r < R - \epsilon$, with $\epsilon$ and $\delta$ related by a relation that is not well understood.

In summary, in a generic configuration entanglement entropies regulated by the UV and geometric cutoff will not match with each other. The point we want to stress is that the mismatching does not indicate that the entanglement entropy regulated by the geometric regulators is not correct. We should not mix between the UV and geometric cutoffs. However, in some special configurations it is meaningful to conduct this matching. This happens when the geometric information in the geometric regulator can be characterized by a single parameter. For example, the spherical regions and infinitely long strips with the width of the cutoff regions being a constant $\epsilon$. See section 6 for an explicit example.

### 3.2 Subtleties in the ALC proposal in higher dimensions

Now we evaluate the entanglement entropy using PEE with a geometric regulator,

$$S_A = \mathcal{I}(A^{reg}, \bar{A}^{reg})|_{A^{reg} \to A, \bar{A}^{reg} \to \bar{A}}. \tag{18}$$

The geometric regularization has the following disadvantages.

- Firstly, the value of the regularized $S_A$ depends on how we take the limit, and furthermore there exists an infinite number of different ways to take the limit.

- Secondly, one can take a similar geometric regularization for $S_{\bar{A}}$. We may expect the existence of a "natural" regularization of $S_{\bar{A}}$ in accordance with the one of $S_A$ thus the universal relation $S_A = S_{\bar{A}}$ can be satisfied. However, this "natural" regularization for $S_{\bar{A}}$ is not obvious to us.

- At last in the *ALC proposal* (12), the left hand side is a finite PEE independent from the regularization scheme, while the right hand side is a linear combination of divergent entanglement entropies need to be regularized. Although the divergent part of the entanglement entropies will cancel with each other either we use the UV regulator or different geometric regulators, the remaining finite terms will indeed depend on the regularization schemes when we use geometric regulators. The validity of (12) needs a suitable choice for the regulators.

In 2-dimensional theories, these disadvantages will not manifest since the boundaries are only points with no geometric information. In higher dimensions, they will severely spoil the validity of the *ALC proposal*. In order to avoid the above disadvantages, we will focus on the configurations satisfying the following requirements:

1. The configuration should have enough symmetries, thus the contour functions only depend on one coordinate.

2. The cutoff region should also respect the symmetries.

3. We will take the suitable regulator for the validity of the *ALC proposal*. More explicitly, the subset entanglement entropies should all be evaluated from the PEE under the same geometric regularization scheme. In other words we should exclude the same class of local partial entanglement while evaluating all the subset entanglement entropies in (12).

The first and second requirements gives a natural order for the degrees of freedom in the system hence eliminate the ambiguities. More importantly this geometric regularization can be characterized by a single parameter, thus the entanglement entropy evaluated under these regulators can possibly match with those regularized by UV cutoff (see section 6). We call these configurations the *quasi-one-dimensional* configurations. The third requirement is substantial for the validity of the universal relation of $S_A = S_{\bar{A}}$ and solves the subtlety of (12).

How can the relation $S_A = S_{\bar{A}}$ be satisfied? We divide the region $A = A^{reg} \cup A^{cut}$, where $A^{cut}$ represents the degrees of freedom near the boundary whose contribution is excluded. Also we conduct a similar decomposition $\bar{A} = \bar{A}^{reg} \cup \bar{A}^{cut}$ for $\bar{A}$. According to the normalization requirements $S_A = s_A(A^{reg})|_{A^{reg} \to A}$, the entanglement entropy $S_A$ (or $S_{\bar{A}}$) can be evaluated by the PEE under the limits $A^{reg} \to A$ (or $\bar{A}^{reg} \to \bar{A}$),

$$S_A = s_A(A^{reg}) = \mathcal{I}(A^{reg}, \bar{A}) = \mathcal{I}(A, \bar{A}) - \mathcal{I}(A^{cut}, \bar{A}), \tag{19}$$

$$S_{\bar{A}} = s_{\bar{A}}(\bar{A}^{reg}) = \mathcal{I}(\bar{A}^{reg}, A) = \mathcal{I}(A, \bar{A}) - \mathcal{I}(\bar{A}^{cut}, A). \tag{20}$$

It is clear that the above regularization for $A$ and $\bar{A}$ excludes different types of short distance PEE, hence in general $S_A \neq S_{\bar{A}}$. More explicitly, in the first case $\bar{A}^{cut}$ is taken to be vanished, while in the second case $A^{cut}$ is vanished.

If we want to keep $S_A = S_{\bar{A}}$, it is more natural to fix the region $A^{cut} \cup \bar{A}^{cut}$ when evaluating the entanglement entropy in both sides, thus

$$S_A = \mathcal{I}(A^{reg}, \bar{A}^{reg}) = \mathcal{I}(\bar{A}^{reg}, A^{reg}) = S_{\bar{A}}. \tag{21}$$

Then the universal relation $S_A = S_{\bar{A}}$ is just a limit of the relation (10), where $A, B, C$ can be taken as $A^{reg}, \bar{A}^{reg}$ and $A^{cut} \cup \bar{A}^{cut}$ respectively. One can take either $A^{cut}$ or $\bar{A}^{cut}$ to be vanished as long as the boundary is contained in $A^{cut} \cup \bar{A}^{cut}$, hence the divergent contribution is removed. Although the entanglement entropy is still scheme dependent, the relation $S_A = S_{\bar{A}}$ will always hold as long as the class of short distance PEE we exclude is fixed.

This requirement also solves the subtlety when applying (12). Decomposing the subsets $\alpha, \alpha_L, \alpha_R$ in (12) into the regularized region and cutoff region, for example $\alpha = \alpha^{reg} \cup \alpha^{cut}$, then the subset entanglement entropies can be written as a PEE, for example,

$$S_{\alpha_L \cup \alpha} = \mathcal{I}(\alpha_L^{reg} \cup \alpha_L^{cut} \cup \alpha^{reg} \cup \alpha^{cut}, \bar{A}^{reg} \cup \bar{A}^{cut} \cup \alpha_R^{reg} \cup \alpha_R^{cut}). \tag{22}$$

Due to the additivity, the above expression can furthermore be decomposed as a summation of the PEEs between smaller divided regions, for example $\mathcal{I}(\alpha_L^{reg}, \bar{A}^{reg})$. Similarly, we decompose the other subset entanglement entropies in (12), then we have

$$\frac{1}{2}\left(S_{\alpha_L \cup \alpha} + S_{\alpha \cup \alpha_R} - S_{\alpha_L} - S_{\alpha_R}\right) = \mathcal{I}(\alpha, \bar{A}), \tag{23}$$

which exactly reproduces the left hand side of (12) which is totally scheme independent with no subtlety when $\alpha$ does not share boundaries with $A$. We summarize that, the validity of (12) requires that, firstly we should evaluate the subset entanglement entropies using geometric regulator; secondly, the cutoff region should be the same while evaluating all the subset entanglement entropies.

So far we do not need the configurations to be *quasi-one-dimensional*. However, in order to apply the *ALC proposal* and compare with the known results with a UV cutoff, we will require the whole configuration, including the theory, regions and regularization scheme, to respect enough symmetries. For example, see the configuration in Fig. 4 with rotation symmetries.

We should always keep in mind that the subset entanglement entropies in the *ALC proposal* should be evaluated by certain limits of the PEEs with the fixed cutoff region. Plugging the entanglement entropies calculated by, for example the RT formula, into (12) will spoil the validity of this proposal.

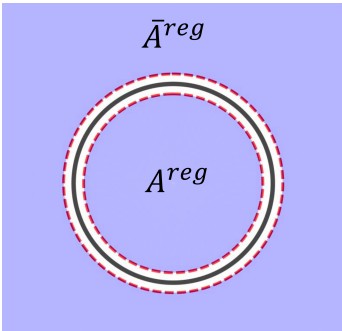

Figure 4: Here both of the region and the geometric regulator respect the rotation symmetries.

### 3.3 Derivative version of the ALC proposal

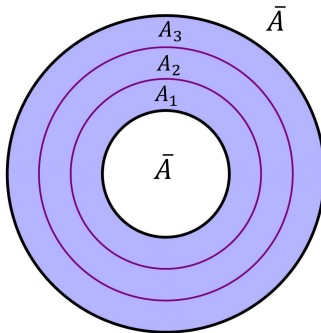

Figure 5: In the above figure we set $A = A_1 \cup A_2 \cup A_3$ to be an annulus whose inner and outer radio is $R_1$ and $R_2$. It is divided into three smaller annulues by the two purple circles.

Here we derive a derivative version of the *ALC proposal*. We take the spherical shells as a typical example for the *quasi-one-dimensional* configurations. For example see Fig. 5, where the cutoff regions do not show up. The whole configuration respects the rotation symmetries, and the *entanglement contour* $s_A(r)$ only depends on the radius coordinate. In order to get the contour function from the *ALC proposal*, we chose $A_2$ to be an infinitely narrow shell that covers the region $r \to r + dr$ with $dr \to 0$. The area of the $(d-2)$-dimensional spherical surface with unit radius is denoted by

$$\Omega_{d-2} = \frac{2\pi^{\frac{d-1}{2}}}{\Gamma\left(\frac{d-1}{2}\right)}. \tag{24}$$

Under the limit $dr \to 0$, according to (12) we have

$$s_A(A_2) = s_A(r)r^{d-2}\Omega_{d-2}dr = \frac{1}{2}\left((S_{A_1 \cup A_2} - S_{A_1}) - (S_{A_3} - S_{A_3 \cup A_2})\right). \tag{25}$$

Since the inner boundary of $A_1$ is fixed, $S_{A_1}$ can be written as a function of the radius of its outer boundary $S_{A_1}(r)$. We can take $S_{A_1 \cup A_2} - S_{A_1}$ as a perturbation of $S_{A_1}$, i.e. $S_{A_1}(r+dr) - S_{A_1}(r)$. Similarly $S_{A_3} - S_{A_3 \cup A_2} = S_{A_3}(r+dr) - S_{A_3}(r)$, where we fixed the outer boundary of $A_3$. Then it is easy to see that,

$$s_A(r) = \frac{1}{2r^{d-2}\Omega_{d-2}}\partial_r\left(S_{A_1}(r) - S_{A_3}(r)\right), \tag{26}$$

which is the derivative version of the *ALC proposal* (12). It has been discussed in [37] for 2-dimensional cases.

## 4 Entanglement contour for spherical regions in CFTs

In this section we derive the *entanglement contour* for spherical regions in general dimensional CFTs in the context of AdS/CFT. The derivation will be carried out with the Rindler method and the fine structure of the entanglement wedge respectively. This contour function is not only interesting by itself, but also the key ingredient for our derivation of the entanglement entropy and *entanglement contour* for spherical shells in the next section.

The *entanglement contour* functions for static spherical regions in $d$-dimensional CFTs has been proposed in an earlier paper [37] based on certain explicit constructions [59] of bit thread configurations [45] on the gravity side[4]. This specific bit thread configuration [59] requires the bit threads to follow the bulk geodesics normal to the RT surface. However, the bit threads configurations are highly non-unique and different configurations will give different contours. Nevertheless, the proposal in [37] coincides with our results.

### 4.1 Entanglement contour from the Rindler method

The Rindler method is developed in [28] to calculate holographic entanglement entropies for static spherical regions in the context of $\text{AdS}_{d+1}/\text{CFT}_d$. Here we focus on the story on the field theory side, where a static spherical region $A$ is considered in the vacuum state of a CFT on a $d$-dimensional hyperplane $B$. The Rindler method constructs the conformal transformation $\mathcal{R}$ that maps the causal development $\mathcal{D}_A$ of $A$ to a Rindler spacetime $\tilde{B}$ with infinitely far away boundary. Since the mapping $\mathcal{R}$ is a symmetry transformation, the entanglement entropy $S_A$ is mapped to the thermal entropy $S_{\tilde{B}}$ of the Rindler spacetime. Furthermore, according to the requirements of PEE, the entanglement structure should also be invariant under $\mathcal{R}$. More explicitly if $\tilde{B}_i$ in $\tilde{B}$ is the image of the subset $A_i$ of $A$ under the Rindler transformation $\mathcal{R}$, then we should have

$$s_A(A_i) = s_{\tilde{B}}(\tilde{B}_i). \tag{27}$$

Taking $A_i$ to be an arbitrary site inside $A$, the above relation gives a unique mapping between the *entanglement contour* function of $\tilde{B}$ and the one of $A$.

The state in the Rindler spacetime $\tilde{B}$ is a thermal state with translation symmetries along all the spatial directions. Again according to the symmetry requirements, the contour function in $\tilde{B}$ is just a constant. So we can directly get the contour function $s_A(r)$ by mapping the flat contour in $\tilde{B}$ back to the region $A$, using the inverse mapping of $\mathcal{R}$[5].

In the following we list the metrics on $B$ and $\tilde{B}$, and the Rindler transformation $\mathcal{R}$[6] between them [28],

$$B: \quad ds^2 = -dt^2 + dr^2 + r^2 d\Omega_{d-2}^2, \tag{28}$$

$$\tilde{B}: \quad ds^2 = -d\tau^2 + R^2 \left( du^2 + \sinh^2 u \, d\Omega_{d-2}^2 \right), \tag{29}$$

$$\mathcal{R}: \quad \left\{ t = R \frac{\sinh(\tau/R)}{\cosh u + \cosh(\tau/R)}, \quad r = R \frac{\sinh u}{\cosh u + \cosh(\tau/R)} \right\}. \tag{30}$$

Here $R$ is the radius of the spherical region $A$. For simplicity we only consider static configurations, thus we set $t = \tau = 0$. The Rindler transformation $\mathcal{R}$ then reduces to,

$$\mathcal{R}: \quad r = R \tanh\left(\frac{u}{2}\right). \tag{31}$$

It is clear from (31) that the region $0 \leq u < \infty$ in $\tilde{B}$ covers the spherical region $0 \leq r < R$, which is just the region $A$ we consider.

Like the entanglement entropy, the thermal entropy $S_{\tilde{B}}$ is also divergent due to the infinite volume of $\tilde{B}$. The next key step of the Rindler method is to regularize $S_{\tilde{B}}$ by regularizing the

---

[4]Actually, the relation between the entanglement contour and the bit thread configuration was first pointed out in a talk by Erik Tonni [60].

[5]The above argument can explain the idea [31, 61] of identifying the inverse of the local weight function, which multiplies the local operator $T_{00}$ in the corresponding modular Hamiltonian $K_A$, as the *entanglement contour* function, i.e., $K_A \propto \int_{x \in A} \frac{T_{00}}{s_A(x)} dx^{d-1}$.

[6]To map from $B$ to $\tilde{B}$, the Rindler transformation $\mathcal{R}$ should be followed by a Weyl transformation that eliminates the overall prefactor of the metric in $\tilde{B}$. We did not write it down because it is spurionic [62] thus does not change the thermal partition function in $\tilde{B}$.

volume of $\tilde{B}$. In other words we only collect the contribution from the region $0 \leq u \leq u_{\max}$ where $u_{\max}$ is the cutoff the volume of $\tilde{B}$. This regulator definitely belongs to the geometric regulator using PEE. The cutoff region in $\tilde{B}$ is mapped to the region $R - \epsilon < r < R$, which is the cutoff region in the original spacetime $B$. According to (31), $u_{\max}$ and $\epsilon$ are related by,

$$R - \epsilon = R \tanh\left(\frac{u_{\max}}{2}\right). \tag{32}$$

Then we work out the contour function $s_A(r)$ based on the flat contour function $s_{\tilde{B}}(u)$ of $\tilde{B}$, which is given by

$$s_{\tilde{B}}(u) = \mathcal{C} = \frac{S_{\tilde{B}}}{Volume(\tilde{B})}. \tag{33}$$

Holographically, the thermal entropy $S_{\tilde{B}}$ equals to the thermal entropy of a $(d+1)$-dimensional hyperbolic black hole[7] [28]

$$ds^2 = \frac{d\rho^2}{\rho^2/L^2 - 1} - (\rho^2/L^2 - 1)d\tilde{\tau}^2 + \rho^2(du^2 + \sinh^2 u\, d\Omega_{d-2}^2), \tag{34}$$

where $L = Re^{\beta}$ is the AdS radius, $\tilde{\tau} = e^{-\beta}\tau$ and $\beta$ is a constant. Using Wald's entropy formula the thermal entropy of the hyperbolic black hole is then given by

$$S_{\tilde{B}} = \frac{c}{6}\int_0^{u_{\max}} du\, (\sinh u)^{d-2}\Omega_{d-2}. \tag{35}$$

Here $\frac{c}{6} = a_d^* \frac{2\Gamma(d/2)}{\pi^{d/2-1}}$ and $a_d^*$ is a central charge that characterizes the number of degrees of freedom in the dual CFT[8] [28, 71]. Since

$$Volume(\tilde{B}) = \int_0^{u_{\max}} du\, R^{d-1}(\sinh u)^{d-2}\Omega_{d-2}, \tag{36}$$

we have

$$\mathcal{C} = \frac{c}{6}\frac{1}{R^{d-1}}. \tag{37}$$

Since the thermal entropy $S_{\tilde{B}}$ equals to the entanglement entropy $S_A$, we have

$$S_A = \int_0^{R-\epsilon} dr\, s_A(r)r^{d-2}\Omega_{d-2} = \mathcal{C}\int_0^{u_{\max}} du\, R^{d-1}(\sinh u)^{d-2}\Omega_{d-2}. \tag{38}$$

Plugging (31) (32) and (37) into the above equation we get the contour function for $(d-1)$-dimensional balls with radius $R$,

$$s_A(r) = \frac{c}{6}\left(\frac{2R}{R^2 - r^2}\right)^{d-1}. \tag{39}$$

---

[7]Here we use holography to calculate the thermal entropy $S_{\tilde{B}}$. The prescription works also for non-holographic CFTs since there are other methods for the evaluation of the thermal entropy. When $d = 2$, $S_{\tilde{B}}$ can be calculated by Cardy-formula without holography. In higher dimensions the thermal entropy can also be evaluated via the heat kernel approach [63], see for example [22, 64]. Furthermore, the Rindler method is also generalized to other 2-dimensional field theories like the warped CFT and field theories invariant under the $BMS_3$ group (BMSFT) in [65, 66] where the thermal entropy of the Rindler space can be calculated using Cardy-like formulas (see also [67–70]) as well as holography.

[8]When $d$ is even, $a_d^*$ is just the coefficient of the A-type trace anomaly in the CFT.

As a consistency check, we consider the $d = 2$ case. According to (39) we have

$$S_A = 2 \int_{r=0}^{r=R-\epsilon} s_A(r) dr = \frac{c}{3} \log \frac{2R}{\epsilon} + \mathcal{O}(\epsilon), \qquad (40)$$

which exactly matches with the entanglement entropy for an interval with length $l = 2R$ evaluated by the RT formula or the replica trick. As we have mentioned before, in 2-dimensional theories, the mixing between the two types of regulators only affects the entanglement entropy at order $\mathcal{O}(\epsilon)$, hence can be omitted. Later we will see that this effect can not be omitted in higher dimensions.

## 4.2 Entanglement contour from the fine structure of the entanglement wedge

In the context of AdS$_3$/CFT$_2$, a holographic picture for the *entanglement contour* of a single interval was given in [25]. It was shown that the entanglement wedge can be sliced by the *modular slices*[9], which are two-dimensional surfaces defined as the orbits of the boundary modular flow lines under the bulk modular flow (the blue surface in the left figure in Fig. 6). Each modular slice intersects with the interval $A$ on a point $P$ and its RT surface $\mathcal{E}_A$ on another point $\tilde{P}$. It was argued that when we apply the replica trick on both of the boundary CFT and the bulk spacetime, cyclically gluing the point $P$ induces a replica story on the corresponding modular slice and turns on the conic defect at $\tilde{P}$, which turns on the nonzero contribution to $S_A$ from $\tilde{P}$ following the Lewkowycz-Maldacena prescription [72]. In other words, for any point $P$ in $A$, its contribution to $S_A$ is represented by its partner point $\tilde{P}$ on $\mathcal{E}_A$. Later we will show that the pair of points $P$ and $\tilde{P}$ are connected by a geodesic normal to $\mathcal{E}_A$. This one-to-one correspondence between the points on $A$ and $\mathcal{E}_A$ perfectly matches our interpretation of the *entanglement contour*. In the same sense, the points in any subinterval $A_i$ in $A$ has partner points forming a geodesic chord $\mathcal{E}_i$ on $\mathcal{E}_A$. So the PEE $s_A(A_i)$ is just given by

$$s_A(A_i) = \frac{Area(\mathcal{E}_i)}{4G}. \qquad (41)$$

See the right figure in Fig. 6 for a graphical description of this correspondence and see [25,26] for more details. Also the similar construction is conducted for WCFT in [36] in the context of AdS$_3$/WCFT correspondence [67,73].

Here we generalize the above construction to static spherical regions $A$ in higher dimensions. For spherical regions, the modular Hamiltonian is local and generates a local geometrical flow, which we call the modular flow. The modular flow in the Rindler spacetime is parametrized by the Rindler time. One can extend the Rindler transformations in the boundary field theory to the bulk. The bulk Rinder transformations will map the entanglement wedge $\mathcal{W}_A$ of $A$ to a hyperbolic black hole $\widetilde{AdS}_{d+1}$ (34), which can be written as a slicing of AdS$_2$ spacetime,

$$\widetilde{AdS}_{d+1} = AdS_2 \times \mathcal{H}_{d-1}. \qquad (42)$$

Here $\mathcal{H}_{d-1}$ is the $(d-1)$-dimensional hyperbolic plane. Under the bulk Rindler transformation the region, $A$ is mapped to a time slice $\tilde{A}$ on the boundary of the $\widetilde{AdS}_{d+1}$, while the RT surface $\mathcal{E}_A$ is mapped to the horizon $\rho = L$ of the hyperbolic black hole, which we denoted by $\tilde{\mathcal{E}}_A$. Since $\widetilde{AdS}_{d+1}$ corresponds to a thermal state, the modular Hamiltonian is just the ordinary Hamiltonian in $\widetilde{AdS}_{d+1}$. In other words the bulk modular flow in $\mathcal{W}_A$ maps to the time evolution

---

[9]Like the Rindler method, the fine structure analysis for the entanglement wedge can also be generalized to the warped CFT and BMSFTs [36].

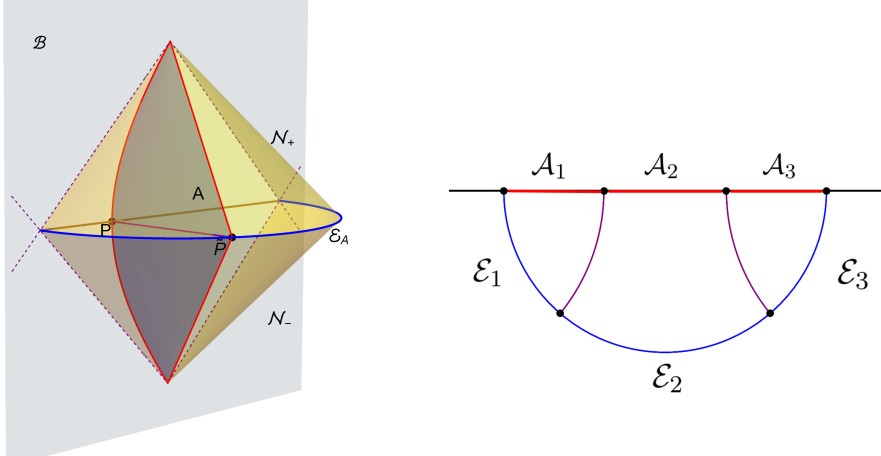

Figure 6: Figures extracted from [54], licenced under CC-BY 4.0. The left figure shows the slicing of the entanglement wedge with modular slices (the blue surface with red boundaries). The right figure shows a time slice of the entanglement wedge and the correspondence between the partial entanglement entropies and geodesic chords on $\mathcal{E}_A$. The purple lines are where the modular slices intersect with the time slice.

in $\widetilde{AdS}_{d+1}$. The modular slices in $\widetilde{AdS}_{d+1}$ are simply the AdS$_2$ slices at a fixed point on the hyperbolic plane $\mathcal{H}_{d-1}$.

The bulk and boundary modular flow in $\mathcal{W}_A$ can be obtained from the Rindler time in $\widetilde{AdS}_{d+1}$ by the inverse Rindler transformations, which is a complicated task. Instead of constructing the modular slices directly in $\mathcal{W}_A$ [25], here we study fine structure in the Rindler $\widetilde{AdS}_{d+1}$ at first. This gives a fine correspondence between the points on $\tilde{A}$ and those on $\tilde{\mathcal{E}}_A$. This is much simpler since the modular slices in $\widetilde{AdS}_{d+1}$ are just the AdS$_2$ slices. Any pair of points $\tilde{A}$ and $\tilde{\mathcal{E}}_A$ have the same coordinates in $\mathcal{H}_{d-1}$. This simple fine correspondence in $\widetilde{AdS}_{d+1}$ is just what maps to the fine correspondence in $\mathcal{W}_A$.

Since we consider the static configurations, we take a time slice for both the original and Rindler spacetime. The modular slices in $\widetilde{AdS}_{d+1}$ intersect with the time slice at lines along the $\rho$ coordinate (see the dashed purple lines in the right figure of Fig. 7). The fine correspondence in $\widetilde{AdS}_{d+1}$ straightforwardly shows that the contour function in $\tilde{A}$ is flat, which is consistent

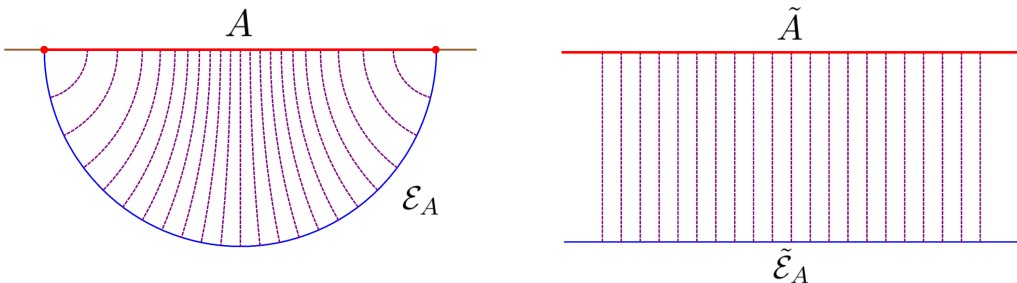

Figure 7: The left figure is extracted from [54], licenced under CC-BY 4.0. Here we have chosen a time slice and fixed all the angular coordinates in the original Poincare AdS$_{d+1}$ as well as the hyperbolic black hole $\widetilde{AdS}_{d+1}$ (34). The dashed purple lines are the intersection between modular slices and the time slice, they are also geodesics normal to the RT surface $\mathcal{E}_A$ and the horizon of the hyperbolic black holes $\tilde{\mathcal{E}}_A$. .

with our previous statement based on the symmetry requirement. These intersection lines can also be determined by the following two properties, firstly they intersect with the horizon $\tilde{\mathcal{E}}_A$ vertically, secondly they are also geodesics in $\widehat{AdS}_{d+1}$. These two properties are kept by their images when mapping back to $\mathcal{W}_A$. In other words, any point $P$ on $A$ and its partner point $\tilde{P}$ on $\mathcal{E}_A$ are connected by static geodesics normal to $\mathcal{E}_A$ (see the dashed purple lines in the left figure of Fig. 7).

We set the radius of the spherical region $A$ to be $R$, its RT surface in Poincaré AdS$_{d+1}$[10] is then given by

$$\mathcal{E}_A: \quad z^2 = R^2 - r^2. \tag{43}$$

Using those static geodesics normal to $\mathcal{E}_A$, we find that the points with radius $r$ in $A$ correspond to the points on the RT surface $\mathcal{E}_A$ with radius $\bar{r}$ via the following relation

$$r = \frac{\bar{r}R}{\sqrt{R^2 - \bar{r}^2} + R}. \tag{44}$$

Note that the above relation is independent from the spacetime dimension $d$.

According to the fine correspondence, we have

$$s_A(r)r^{d-2}\Omega_{d-2}dr = \frac{1}{4G}\sqrt{\frac{R^2\bar{r}^{2(d-2)}}{(R^2 - \bar{r}^2)^d}}\Omega_{d-2}d\bar{r}, \tag{45}$$

where the left-hand side is the PEE of a thin annulus at $r$ with the width $dr$ while the right hand side is the area of the subregion on the RT surface corresponding to that annulus. Plugging the fine correspondence relation (44) into the above equation, we immediately get

$$s_A(r) = \frac{1}{4G}\left(\frac{2R}{R^2 - r^2}\right)^{d-1}, \tag{46}$$

which reproduces the result (39) from the Rindler method.

## 5 Entanglement entropy and entanglement contour for annuli and spherical shells

Again, in this section we consider the vacuum state of a CFT$_d$ on a hyperplane with Poincaré symmetry. Based on the contour function for spherical regions (39) and the *ALC proposal*, in this section we derive the *entanglement contour* and entanglement entropy for annulus and spherical shells in CFT$_d$. Firstly we consider the region $A$ to be a disk and its partition into a concentric smaller disk $A_1$ and an annulus $A_2$, see Fig. 8. Note that, since there are only two relevant subsets, the *ALC proposal* (12) reduces to the following equation,

$$S_{A_2} = S_A + S_{A_1} - 2s_A(A_1). \tag{47}$$

Our strategy to derive the entanglement entropy for annuli is the following

1. Calculate the partial entanglement entropies $s_A(A_1)$ and evaluate the entanglement entropies $S_A$ and $S_{A_1}$ based on the contour function for the spherical region (39). When evaluating $S_A$ (or $S_{A_1}$), we integrate the contour function from $r = 0$ to a radius that is slightly smaller than the radius of $A$ (or $A_1$). This corresponds to the choice of the cut-off region shown in Fig. 8. Furthermore we fixed the width of all the cutoff regions to be an infinitesimal constant $\epsilon$.

---

[10]Here we use the following metric $ds^2 = \frac{1}{z^2}\left(-dt^2 + dr^2 + r^2d\Omega_{d-2}^2 + dz^2\right)$ where $z$ is the radius coordinate in the bulk.

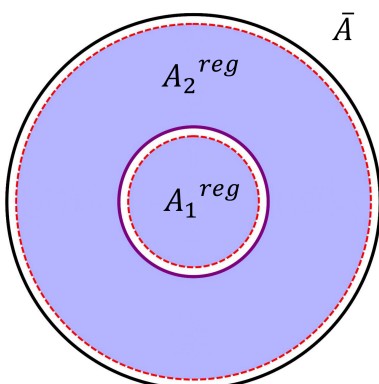

Figure 8: We consider a disk region $A$ which is divided into a concentric smaller disk and an annulus $A = A_1 \cup A_2$. The white region bounded by the solid circle boundaries and the red dashed circles are the cutoff regions when we evaluate the entanglement entropies $S_A$ and $S_{A_1}$ in the *ALC proposal*. The cutoff region for $\bar{A}$ is taken to be empty.

2. Evaluate the entanglement entropy of the annulus $S_{A_2}$ using (47). Since the validity of the *ALC proposal* requires the cutoff region to be the same, the $S_{A_2}$ we get in such a way is also regularized by the cutoff regions shown in Fig. 8.

3. Then we consider another configuration where an annulus $A$ is subdivided into two smaller annuli $A_1$ and $A_2$, see Fig. 9. In order to use the entanglement entropy for annulus we get in the previous step, we choose the cutoff regions at all the boundaries as in Fig. 9. More explicitly, for either of the two annuli the cutoff region at the outer boundary is inside the annulus, while the cutoff region at the inner boundary is outside the annulus, which is the same as Fig. 8. Plugging the entanglement entropy for annuli into the derivative version of the *ALC proposal*, then we get the *entanglement contour* function for annuli.

4. Based on the contour function for annuli, we evaluate the entanglement entropy for annuli with other choice of the cutoff regions.

Let us begin with Fig. 8. We set the radius of $A$ to be $R_2$ and the radius of $A_1$ to be $R_1$. The *entanglement contour* function for disks is given by (39) with $d = 3$. So we can easily get the regularized entanglement entropy for disks $A$ and $A_2$,

$$S_A = \int_0^{R_2-\epsilon} \frac{c}{6} \left( \frac{2R_2}{R_2^2 - r^2} \right)^2 2\pi r \, dr = \frac{2\pi c}{3} \frac{(R_2 - \epsilon)^2}{\epsilon(2R_2 - \epsilon)},$$

$$S_{A_1} = \int_0^{R_1-\epsilon} \frac{c}{6} \left( \frac{2R_1}{R_1^2 - r^2} \right)^2 2\pi r \, dr = \frac{2\pi c}{3} \frac{(R_1 - \epsilon)^2}{\epsilon(2R_1 - \epsilon)}. \tag{48}$$

The PEE $s_A(A_1)$ is calculated by

$$s_A(A_1) = \mathcal{I}(A_1, \bar{A}) = \int_0^{R_1} \frac{c}{6} \left( \frac{2R_2}{R_2^2 - r^2} \right)^2 2\pi r \, dr = \frac{2\pi c R_1^2}{3\left(R_2^2 - R_1^2\right)}. \tag{49}$$

Then according to (47) we can straightforwardly get the entanglement entropy of any annulus with outer and inner radius being $R_2$ and $R_1$ respectively,

$$S_{annulus} = \frac{2\pi c}{3} \left( \frac{(R_2 - \epsilon)^2}{(2R_2 - \epsilon)\epsilon} + \frac{(R_1 - \epsilon)^2}{(2R_1 - \epsilon)\epsilon} - \frac{2R_1^2}{R_2^2 - R_1^2} \right). \tag{50}$$

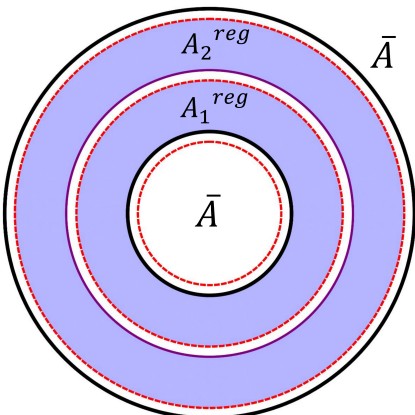

Figure 9: The partition of an annulus $A = A_1 \cup A_2$ with rotation symmetry. The white thin annuli are the cutoff regions at each boundary. The width of all the cutoff annuli is set to be $\epsilon$.

Note again that, the above result is evaluated under the special choice of the cutoff region shown Fig. 8.

Then we turn to the configuration in Fig. 9 where $A$ changes to be an annulus. Again we denote the outer and inner radius of $A$ to be $R_2$ and $R_1$. We divide $A$ into two thinner annuli $A_1$ and $A_2$ by a circle $r = R$ which satisfies $R_1 \le R \le R_2$. Since we will use the formula (50) to evaluate the entanglement entropies for the subsets $A_1$ and $A_2$, the cutoff regions have been properly set and fixed as in Fig. 9. Again all the cutoff regions are narrow annuli with the same width $\epsilon$. Under this configuration we immediately get,

$$S_{A_1} = \frac{2\pi c}{3} \left( \frac{2R_1^2}{R_1^2 - R^2} + \frac{(R-\epsilon)^2}{(2R-\epsilon)\epsilon} + \frac{(R_1-\epsilon)^2}{(2R_1-\epsilon)\epsilon} \right), \tag{51}$$

$$S_{A_2} = \frac{2\pi c}{3} \left( \frac{2R^2}{R^2 - R_2^2} + \frac{(R_2-\epsilon)^2}{(2R_2-\epsilon)\epsilon} + \frac{(R-\epsilon)^2}{(2R-\epsilon)\epsilon} \right). \tag{52}$$

Since $R_1$ and $R_2$ are fixed, $S_{A_1}$ and $S_{A_2}$ are functions of $R$. Then we plug the above equations into (26) and get the contour function for a general annulus with its outer and inner radius being $R_2$ and $R_1$,

$$s_{annulus}(r) = \frac{1}{4\pi r} \partial_R \left( S_{A_1} - S_{A_2} \right)|_{R \to r} = \frac{2c}{3} \left( \frac{R_2^2}{\left( r^2 - R_2^2 \right)^2} + \frac{R_1^2}{\left( r^2 - R_1^2 \right)^2} \right). \tag{53}$$

We want to stress that, unlike the entanglement entropy the contour function is scheme independent. We will always get the contour function (53) as long as the cutoff regions are fixed while applying the *ALC proposal*. As was checked in a later paper [27], the PEE (49) coincides with the one evaluated by the EMI formula (15). The results (50) and (53) for the vacuum state of CFT are derived based on (49) and the *ALC proposal* which is also consistent with the EMI formula. One can also check that they can be reproduced by the EMI formula.

We can take two interesting limits. Firstly, when the inner boundary $R_1 \to 0$, the annulus becomes a disk. As expected the contour function (56) with $R_1 = 0$ recovers the contour function for spherical regions (39) with $d = 3$. Secondly, let us set $R_2 = R_1 + L$ and take the limit $R_1 \to \infty$ while keeping $L$ finite. Under this limit the annulus becomes a strip of width $L$, and the contour function (53) gives the contour function for strips. Let us define $x = R_1 + \frac{L}{2}$

hence $x = 0$ locates at the center of the strip, the contour function is then given by,

$$s_{strip}(x) = \frac{2c}{3}\left(\frac{1}{(L-2x)^2} + \frac{1}{(L+2x)^2}\right). \tag{54}$$

Using the *entanglement contour* function (53), we can also evaluate the entanglement entropy for annuli under other regularization schemes. We consider a regularized region

$$A^{reg} : \{R_1 + \epsilon_1 \le r \le R_2 - \epsilon_2\}, \tag{55}$$

and calculate its contribution to $S_A$. In other words we calculate the PEE $s_A(A^{reg}) = \mathcal{I}(A^{reg}, \bar{A})$, with the cutoff region in $\bar{A}$ vanishes. This is different from our previous schemes. Under the limit $\epsilon_1, \epsilon_2 \to 0$, we get the regularized entanglement entropy for annuli,

$$S_A = s_A(A^{reg}) = \int_{R_1+\epsilon_1}^{R_2-\epsilon_2} 2\pi r s_A(r) dr \tag{56}$$

$$= \frac{2\pi c}{3}\left[\frac{R_2^2}{2R_2\epsilon_2 - \epsilon_2^2} - \frac{R_2^2}{R_2^2 - (R_1+\epsilon_1)^2}\frac{R_1^2}{2R_1\epsilon_1 + \epsilon_1^2} - \frac{R_1^2}{(R_2-\epsilon_2)^2 - R_1^2}\right].$$

Expanding the above result with respect to $\epsilon_{1,2}$, we get

$$S_A = \frac{c}{6}\left(\frac{2\pi R_2}{\epsilon_2} + \frac{2\pi R_1}{\epsilon_1} - 4\pi\frac{(R_2^2 + R_1^2)}{(R_2^2 - R_1^2)} + \mathcal{O}(\epsilon_1) + \mathcal{O}(\epsilon_2)\right), \tag{57}$$

which satisfies the area law and has a universal $\mathcal{O}(1)$ term independent from the choice of $\epsilon_1$ and $\epsilon_2$.

It is easy to extend the above discussion to higher dimensions. Based on the contour function (39) for balls in general dimensions, we can evaluate the entanglement entropy for a $(d-1)$-dimensional spherical region $A$ with radius $R_2$ in CFT$_d$,

$$S_A = \int_0^{R_2-\epsilon} s_A(r)\Omega_{d-2}r^{d-2}dr$$

$$= \frac{c}{6}\pi^{\frac{d-1}{2}}\left(2 - \frac{2\epsilon}{R_2}\right)^{d-1} {}_2\tilde{F}_1\left(\frac{d-1}{2}, d-1; \frac{d+1}{2}; \frac{(R_2-\epsilon)^2}{R_2^2}\right), \tag{58}$$

where ${}_2\tilde{F}_1(a, b, c, z) = {}_2F_1(a, b; c; z)/\Gamma(c)$ is the regularized hypergeometric function and $\epsilon \to 0$ is geometric cutoff. We consider $A_1$ as a smaller concentric spherical region with radius $R_1 < R_2$, and $A_2$ as the spherical shell characterized by $R_1 < r < R_2$. Then $S_{A_1}$ is also given by (58) with $R_2$ replaced by $R_1$. The PEE for $s_A(A_1)$ is given by

$$s_A(A_1) = \int_0^{R_1} s_A(r)\Omega_{d-2}r^{d-2}dr$$

$$= \frac{c}{6}\pi^{\frac{d-1}{2}}\left(\frac{2R_1}{R_2}\right)^{d-1} {}_2\tilde{F}_1\left(\frac{d-1}{2}, d-1; \frac{d+1}{2}; \frac{R_1^2}{R_2^2}\right). \tag{59}$$

According to (47) we get the $S_{A_2}$ under the regularization scheme shown by a higher dimensional extension of Fig. 8,

$$S_{A_2} = \frac{c}{6}\pi^{\frac{d-1}{2}}\left[\left(2 - \frac{2\epsilon}{R_2}\right)^{d-1} {}_2\tilde{F}_1\left(\frac{d-1}{2}, d-1; \frac{d+1}{2}; \frac{(R_2-\epsilon)^2}{R_2^2}\right)\right.$$

$$+ \left(2 - \frac{2\epsilon}{R_1}\right)^{d-1} {}_2\tilde{F}_1\left(\frac{d-1}{2}, d-1; \frac{d+1}{2}; \frac{(R_1-\epsilon)^2}{R_1^2}\right)$$

$$\left. - 2^d\left(\frac{R_1}{R_2}\right)^{d-1} {}_2\tilde{F}_1\left(\frac{d-1}{2}, d-1; \frac{d+1}{2}; \frac{R_1^2}{R_2^2}\right)\right]. \tag{60}$$

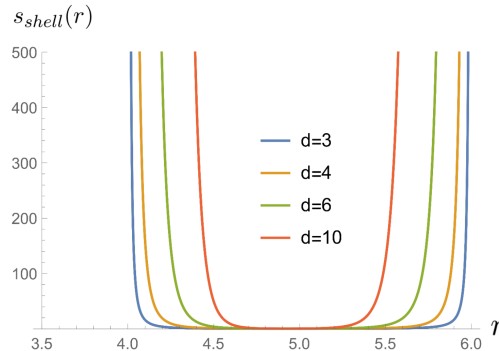

Figure 10: This figure shows the *entanglement contour* function $s_{shell}(r)$ for spherical shells with $c = 1, R_1 = 4, R_2 = 6$ in $d = 3, 4, 6, 10$ dimensional spacetime.

Then we consider the configuration as a higher dimensional generalization of Fig. 9. Applying the similar strategy, we can calculate the *entanglement contour* function for spherical shells with the inner and outer radius being $R_1$ and $R_2$ via the derivative version of the *ALC proposal* (26),

$$s_{shell}(r) = \frac{2^{d-2}c}{3}\left(\left(\frac{R_2}{R_2^2 - r^2}\right)^{d-1} + \left(\frac{R_1}{r^2 - R_1^2}\right)^{d-1}\right). \tag{61}$$

The above result is obviously consistent with our previous result (53) when $d = 3$. Fig. 10 shows the contour functions for spherical shells in different dimensions.

## 6 Comparison between the UV and geometric regulators

As we mentioned previously, the entanglement entropy can be evaluated as a limit of PEE with geometric regulators, which are totally different from the UV regulators. It is quite interesting to see how these two kinds of schemes are related to each other. The UV cutoff is usually captured by a single parameter $\delta$ that labels the scale where we stop counting the entanglement. While the geometric cutoff contains much more information about the geometry of the boundary and cutoff region, which is described by more than one parameter. In general these two schemes cannot be identified by any relations, unless in the special cases where the geometric information can be captured by one parameter. These include the *quasi-one-dimensional* configurations with the cutoff regions also respecting the symmetries. In the following, we will consider the match between these two kinds of schemes for spherical regions and annuli. We succeed for the spherical regions but fail for the annuli.

### 6.1 An exact relation between the UV and geometric cutoff for spherical regions

The fine structure of the entanglement wedge gives us a perfect tool to establish the relation between these two regularization schemes. According to the fine structure we established previously, the contribution from any point $P$ in the spherical region is given by its partner point $\tilde{P}$ on the RT surface. The $z$ coordinate of $\tilde{P}$ furthermore implies that this contribution from $P$ can be effectively taken as a contribution from a definite energy scale. In other words, the contour function in the spatial space indeed corresponds to a contour function in the energy-momentum space due to this fine correspondence. This is quite special. If we let $P$ approach the boundary with an infinitesimal distance $\epsilon$, the $z$ coordinate of its partner point $\tilde{P}$ will also approach an infinitesimal value $\delta$, which is related to $\epsilon$ under the fine correspondence.

Imposing a geometric cutoff for the spherical region at the radius coordinate with $\epsilon$ equates to imposing a UV cutoff at $\delta$.

Then we work out the explicit relation between $\delta$ and $\epsilon$. In the case of static spherical region with radius $R$, let us consider the points with $r = R - \epsilon$ in the spherical region, according to the fine correspondence (44) we have

$$R - \epsilon = \frac{\bar{r}R}{\sqrt{R^2 - \bar{r}^2} + R}, \qquad \bar{r} = \sqrt{R^2 - \delta^2}, \tag{62}$$

where $\bar{r}$ is the radius coordinate of the partner points on the RT surface with the $z$ coordinate denoted as $\delta$. This gives an exact relation between the geometric and UV cutoffs

$$\epsilon = \frac{R\left(R + \delta - \sqrt{(R^2 - \delta^2)}\right)}{R + \delta} = \delta - \frac{\delta^2}{2R} + \frac{\delta^3}{2R^2} - \frac{3\delta^4}{8R^3} + \frac{3\delta^5}{8R^4} + O\left(\delta^6\right). \tag{63}$$

Note that the above relation is independent of $d$ hence works in general dimensions.

At the leading order $\epsilon \sim \delta$, hence the leading area contribution takes the same formula in both of the two regularization schemes. Also in two dimensional theories, the entanglement entropy using the two schemes have the same formula because the difference in sub-leading order only affects the entanglement entropy at the order $\mathcal{O}(\epsilon)$. This justifies the validity of the *ALC proposal* in 2-dimensional theories in the previous literature, by directly plugging into the proposal with the subset entanglement entropies calculated with a UV cutoff.

However, as we mentioned before, in higher dimensions the entanglement entropy evaluated by the Rindler method is expected to deviate from the one evaluated from the RT formula at all orders except the leading one. The Rindler method has been quite extensively studied in the previous literature, including several important developments about our understanding of the holographic entanglement entropy and have a great impact in the community. For example the first derivation of the RT formula for some special cases [28] in AdS/CFT and the original extension of the RT formula to holography beyond AdS/CFT [65] [66]. The deviation also happens to those entanglement entropies [41–43] evaluated from the EMI formula (15) [27, 38].

The concept of the *entanglement contour* or the PEE is relatively new to the community, while the Rindler method and the EMI formula are proposed before the study of the *entanglement contour*. Subtleties about the deviation have already shown up in many papers [22, 28, 38, 43]. When $d > 2$, then entanglement entropy generically has the following expansion

$$S_A = c_{d-2}\left(\frac{R}{\delta}\right)^{d-2} + c_{d-3}\left(\frac{R}{\delta}\right)^{d-3} + c_{d-4}\left(\frac{R}{\delta}\right)^{d-4} + \cdots \begin{cases} (-1)^{\frac{d}{2}-1} 4c_0 \log \frac{R}{\delta}, & \text{even } d, \\ (-1)^{\frac{d-1}{2}} 2\pi c_0, & \text{odd } d, \end{cases} \tag{64}$$

where $R$ characterize the size of the region, $c_i$ are constants and $\delta$ is the cutoff. If one naively takes the geometric cutoff $\epsilon$ as the UV cutoff $\delta$, then the entanglement entropy evaluated under the two different regularization schemes will deviate at all orders except the leading one. The deviation at the universal term is even more disturbing because it was proposed to give the $c$-function of the theory [71, 74]. It is realized in Ref. [22] that the cutoff should be modified in a certain way [57] in order to match the universal term from the Rindler method to the one from the holographic calculation. However the physical meaning behind this modification was not well understood and the prescription can only reproduce the universal term rather than the entanglement entropy in $d > 3$.

In this paper, we clarify the difference between the geometric and UV cutoff. Furthermore, we build the exact relation (63) between them for spherical regions in CFTs in the context of *entanglement contour* and its holographic picture. Plugging (63) into the entanglement

entropy (58) (with $R_2$ replaced by $R$) for spherical regions, we get exactly the holographic entanglement entropy for spheres with radius $R$,

$$S_{sphere}^{hol} = \frac{c\pi^{\frac{d-1}{2}}\left(\frac{R^2}{\delta^2}-1\right)^{\frac{d-1}{2}}{}_2F_1\left(\frac{1}{2},\frac{d-1}{2};\frac{d+1}{2};1-\frac{R^2}{\delta^2}\right)}{3(d-1)\Gamma\left(\frac{d-1}{2}\right)}. \tag{65}$$

This gives a perfect interpretation for the deviation.

## 6.2 Exploration for annulus

In this subsection we compare (56) with the holographic entanglement entropy calculated by the RT formula [75–79]. Again let us consider the region $A_2$ in Fig. 8, there are two minimal surfaces for the annulus.

- Two-disk phase: the extremal surface is disconnected and is the union of the RT surfaces of the two disk $A$ and $A_2$, i.e., $\mathcal{E}_{A_2} = \mathcal{E}_A \cup \mathcal{E}_{A_1}$.

- Hemi-torus phase: the extremal surface is a connected surface of a hemi-torus. The area of this surface is calculated in [75, 79].

The holographic entanglement entropy $S_{A_2}$ is then given by the minimal surface with smaller area. Note that both of the two areas are infinite, it does not make sense to compare their size unless they are regularized. The standard regularization scheme used in the RT formula is to take a cutoff at $z = \delta$ in the AdS bulk. Note that this is not the geometric regularization we discussed in this paper. Under this scheme if we adjust the ratio $R_2/R_1$, the minimal surfaces switch between these two phases at a critical value of the ratio $R_2/R_1 \approx 2.4$. When $R_2/R_1$ is larger than the critical value, the minimal surface will be in the two-disk phase.

In the two-disk phase the entanglement entropy $S_{A_2}^{h1} = S_A^{holo} + S_{A_1}^{holo}$, which expands as

$$S_{A_2}^{h1} = \frac{c}{6}\left(\frac{2\pi R_1}{\delta} + \frac{2\pi R_2}{\delta} - 4\pi + \mathcal{O}(\delta)\right). \tag{66}$$

Note that the MI crossing the annulus vanishes in this phase,

$$I(A_1, \bar{A}) = 0. \tag{67}$$

In the hemi-torus phase the holographic entanglement entropy has the following expansion [75, 76, 78, 79]

$$S_{A_2}^{h2} = \frac{c}{6}\left(\frac{2\pi R_1}{\delta} + \frac{2\pi R_2}{\delta} - \frac{4\pi}{\sqrt{2\kappa^2-1}}\left(\mathbb{E}(\kappa^2) - (1-\kappa^2)\mathbb{K}(\kappa^2)\right) + \mathcal{O}(\delta)\right), \tag{68}$$

where $\kappa$ is a constant determined by the ratio $R_2/R_1$

$$\log\frac{R_1}{R_2} = 2\kappa\sqrt{\frac{1-2\kappa^2}{\kappa^2-1}}\left(\mathbb{K}\left(\kappa^2\right) - \Pi\left(1-\kappa^2|\kappa^2\right)\right), \tag{69}$$

and $\mathbb{E}, \mathbb{K}$ are incomplete or complete elliptic integrals[11]. If we naively take $\epsilon_1 = \epsilon_2 = \epsilon = \delta$, then the entanglement entropy (57) with a uniform geometric cutoff differs from both holographic entanglement entropies (66) and (68) at the universal term. This is expected as we

---

[11]The incomplete elliptic integrals of the first, second and third kind are defined respectively by

$$\mathbb{F}(x|m) \equiv \int_0^x \frac{d\theta}{\sqrt{1-m\sin^2\theta}}, \tag{70}$$

$$\mathbb{E}(x|m) \equiv \int_0^x \sqrt{1-m\sin^2\theta}\,d\theta, \tag{71}$$

$$\Pi(n,x|m) \equiv \int_0^x \frac{d\theta}{(1-n\sin^2\theta)\sqrt{1-m\sin^2\theta}}. \tag{72}$$

stressed that the geometric regulator and the UV regulator exclude different types of correlations.

Since we choose the geometric regularization scheme that respects the symmetries, the geometric cutoff $\epsilon$ could be related to the cutoff $\delta$ in some way, as in the case of the spherical regions that we previously discussed. However, this relation in the case of annulus is more complicated. Firstly, the modular flow is not known, hence we do not know how to conduct the fine structure analysis of the entanglement wedge. The exact relation between the two kinds of cutoff cannot be explored following the prescription in the spherical region cases. Secondly, the two boundaries of the annulus are not symmetric, which means that the cutoff relations for the outer and inner boundaries are different.

For simplicity we consider the case of stripes in $d = 3$, where the second complication is avoided and the relation between $\delta$ and $\epsilon$ should be the same for both of the boundaries. Integrating the contour function (54) from $-\frac{l}{2}+\epsilon$ to $\frac{l}{2}-\epsilon$, we get the geometrically regularized entanglement entropy for strips with width $l$,

$$S_{strip} = \frac{c}{3}L\left(\frac{1}{\epsilon-l}+\frac{1}{\epsilon}\right) = \frac{c}{3}L\left(\frac{1}{\epsilon}-\frac{1}{l}+O(\epsilon)\right), \tag{74}$$

where $L$ is the infinite volume along the extensive direction and $\epsilon$ is the geometric cutoff on both of the two boundaries. On the other hand, setting $\delta$ to be the UV cutoff on both of the boundaries, we get the holographic entanglement entropy

$$S_{strip}^{hol} = \frac{c}{3}L\left(\frac{1}{\delta}-\frac{2\pi\Gamma\left(\frac{3}{4}\right)^2}{\Gamma\left(\frac{1}{4}\right)^2}\frac{1}{l}\right). \tag{75}$$

If the two results match with each other, the relation between $\epsilon$ and $\delta$ should have the following expansion

$$\epsilon = \delta + \left(\frac{2\pi\Gamma\left(\frac{3}{4}\right)^2}{\Gamma\left(\frac{1}{4}\right)^2}-1\right)\frac{\delta^2}{l}+\mathcal{O}\left(\delta^3\right). \tag{76}$$

Then we give two candidate prescriptions to determine the relation between $\epsilon$ and $\delta$ similar to the case of the spherical regions. Due to the translational symmetry along the extensive direction, we can fix the extensive coordinate and consider a cross section of the entanglement wedge, hence the RT surface reduces to a curve in the $(z, x)$ plane. The first prescription is to use the geodesics in AdS that are normal to the RT surface to determine the fine correspondence. More explicitly, we consider a geodesic emanating vertically from some point with $z = \delta$ on the curve, this geodesic will intersect with the boundary at $x = \frac{l}{2}-\epsilon$. Then we get a relation between $\epsilon$ and $\delta$ that has the following expansion

$$\epsilon = \delta - \frac{\pi\Gamma\left(\frac{3}{4}\right)^2}{6\Gamma\left(\frac{5}{4}\right)^2}\frac{\delta^3}{l^2}+\mathcal{O}\left(\delta^5\right), \tag{77}$$

which is different from the expected relation (76). Another disadvantage for the geodesic prescription is that it is not possible to build a family of non-intersecting geodesics normal to the RT surface of the strip when $d > 3$ [59].

---

And the complete elliptic integrals of the first, second and third kind are given by

$$\mathbb{K}(m) = \mathbb{F}(\frac{\pi}{2}|m), \qquad \mathbb{E}(m) = \mathbb{E}(\frac{\pi}{2}|m), \qquad \Pi(n, m) = \Pi(n, \frac{\pi}{2}|m). \tag{73}$$

The second prescription is to use the curves as the cross-section of the RT surfaces of strips to establish the fine correspondence. Note that these cross-section curves are not geodesics in the bulk. Again these curves are also required to be normal to the RT surface of the strip we consider. Then we find the following relation

$$\epsilon = \frac{\sqrt{\pi}\Gamma\left(\frac{7}{4}\right)}{3\Gamma\left(\frac{5}{4}\right)}\delta - \frac{2\pi\Gamma\left(\frac{3}{4}\right)^2}{3\Gamma\left(\frac{1}{4}\right)^2}\frac{\delta^3}{l^2} + \mathcal{O}\left(\delta^5\right). \tag{78}$$

Here $\delta$ and $\epsilon$ differ even at the leading order, hence is obviously inconsistent with the expected relation (76).

More details about how the vertical geodesics or cross-section curves of the RT surfaces can be found in Ref. [59]. However the inconsistency between (76) and (77)(78) implies that neither of the two prescriptions gives the expected fine correspondence between the two cutoffs.

## 6.3 Geometric regulator via the reflected entropy and the PEE

As we mentioned in section 1, the reflected entropy is also a natural candidate to regulate the entanglement entropy. The reason is that when the complement of $A \cup B$ becomes infinitesimal, the area of the EWCS $\Sigma_{AB}$ that holographically dual to the (half of the) reflected entropy, approaches the RT surface $\mathcal{E}_A$ with a cutoff where it anchored on disconnected RT surface $\mathcal{E}_{A \cup B}$. Here we show that this configuration is indeed equivalent to a geometric regulation via the PEE. The key point which support this equivalence is that the EWCS intersect with $\mathcal{E}_{A \cup B}$ vertically, hence $\mathcal{E}_{A \cup B}$ can be considered as the geodesic connecting a pair of points related by the fine correspondence. In other words, the EWCS corresponds to a PEE. The explicit relation between the EWCS and the PEE is studied in [54].

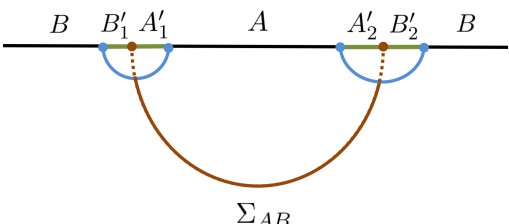

Figure 11: The extension of the EWCS $\Sigma_{AB}$ intersect with the boundary at two points $P_1$ and $P_2$ which divide the complement of $A \cup B$ into $A'_1 \cup B'_1$ and $A'_2 \cup B'_2$.

It was shown in [54] that in the canonical purification of the mixed state $\rho_{AB}$ the reflected entropy equals to the so called balanced entanglement entropy (BPE),

$$\frac{1}{2}S_R(A,B) = \text{BPE}(A,B). \tag{79}$$

Also in the case of Fig. 11 we have

$$\frac{1}{2}S_R(A,B) = \text{BPE}(A,B) = \mathcal{I}(A,B \cup B') = \mathcal{I}(B,A \cup A'). \tag{80}$$

Here $A' = A'_1 \cup A'_2$ and $B' = B'_1 \cup B'_2$ are disconnected regions, and the partition (the position of the two brown partition points) of the complement of $A' \cup B'$ is determined by the following balance condition

$$s_{AA'}(A'_1) = s_{BB'}(B'_1), \qquad s_{AA'}(A) = s_{BB'}(B). \tag{81}$$

The above requirements are called the balance requirements. We also have $s_{AA'}(A'_2) = s_{BB'}(B'_2)$ regarding the fact that the total system is in a pure state. When $A' \cup B'$ approaches the empty space, according to (80) the BPE($A, B$) will approach the entanglement entropy $S_A$ or $S_B$ with the cutoff region settled to be $A'$ or $B'$.

In the holographic case of Fig. 11, the position of the partition points are just where the extension of $\Sigma_{AB}$ anchored at the boundary. The equivalence between the BPE and the reflected entropy goes beyond the holographic cases. More importantly the BPE can be defined for generic purifications (like the one in Fig. 11) hence can be considered as a generalization of the reflected entropy beyond the canonical purification.

# 7 Conclusion

We give a classification to the geometric regulators under which we can evaluate the entanglement entropies by taking certain limits for a well-defined information theoretical quantity. The quantity we used in this paper is the *partial entanglement entropy*. We showed that the geometric regulators exclude different types of correlations from the UV regulators, including both of the short and long distance correlations. This hence implies that the inconsistency between entropies evaluated by the two types of regulators is expected in general configurations. They only match at the leading order. In $d \geq 3$ where the sub-leading terms in the expansion of entanglement entropy do not vanish, the mixing between the regulators will cause severe confusion. The entanglement entropy with a geometric regulator is very sensitive to the cutoff region we chose, which usually include geometric information that cannot be characterized by a single parameter. This sensitivity is expected and should not be understood as an ambiguity of the entanglement entropy under the geometric regulators.

Like the MI, the PEE is free from divergence and satisfies the property of normalization. Furthermore, the additional property of additivity makes the PEE a natural quantity to evaluate the entanglement entropy under a geometric regulator. We demonstrated that the evaluation of the entanglement entropy using the Rindler method or taking certain limits for the EMI and the reflected entropy are indeed examples of evaluating entanglement entropy using the PEE with a geometric regulator. On the other hand the Rindler method and the EMI formula are also useful to derive the *entanglement contour* in general dimensional CFTs.

The generalization of the *ALC proposal* to higher dimensions for *quasi-one-dimensional* configurations is a natural step to consider, since all the reasons for the holding of the proposal [25, 26, 37] in 2 dimensions still hold in higher dimensions. However, there is an important subtlety, which is crucial for the validity of the proposal in higher dimensions. Unlike the 2-dimensional configurations, the subset entanglement entropies in the *ALC proposal* should all be evaluated as a limit of the PEE with a fixed cutoff region. Naively plugging the entanglement entropies calculated with a UV cutoff into the *ALC proposal* will spoil the validity of the proposal. Though the power of the *ALC proposal* is limited by this subtlety in higher dimensions, it can still be useful. For example, based on the contour function for spherical regions and the *ALC proposal*, we derived the contour function for annuli and spherical shells in the vacuum state of a CFT in general dimensions.

Another subtlety arises when we consider the MI across an annulus. For example, we consider the configuration in Fig. 8 and assume that the total system is in a pure state, thus $S_{A_2} = S_{\bar{A} \cup A_1}$. Following the *ALC proposal*, the linear combination that gives the PEE $\mathcal{I}(A_1, \bar{A})$ coincide with (half of) the MI,

$$\mathcal{I}(A_1, \bar{A}) = \frac{1}{2}\left(S_A + S_{A_1} - S_{A_2}\right) = \frac{1}{2}\left(S_{\bar{A}} + S_{A_1} - S_{A_1 \cup \bar{A}}\right) = \frac{1}{2}I(A_1, \bar{A}). \tag{82}$$

Since the contour function $s_A(r)$ for the disk $A$ is non-vanishing everywhere the linear combi-

nation for $\mathcal{I}(A_1, \bar{A})$ should always be positive for any value of the ratio $R_2/R_1$. On the other hand it is well known that the MI $I(A_1, \bar{A})$ vanishes when the ratio $R_2/R_1$ is larger than the critical value. Then we come up with a puzzle that the same linear combination of entanglement entropies will have different values, that depends on whether it is interpreted as the PEE or the MI. Naively plugging into the holographic entanglement entropies into the *ALC proposal* will give us the MI. While the PEE is evaluated under some suitable geometric regulators. An important lesson we can learn from this is that, any quantity written as a linear combination of divergent entanglement entropies should also address the corresponding regularization scheme for the entanglement entropies. The quantity is determined by both the linear combination and the regularization scheme.

The fine structure analysis of the entanglement wedge gives a holographic picture for the *entanglement contour*. For the spherical regions in holographic CFTs, this picture effectively relates the contribution to $S_A$ from a certain site in $A$ to the contribution from a certain scale. In these *quasi-one-dimensional* cases with a cutoff region respecting the symmetries, the geometric information of the geometric regulator indeed can be characterized by a single parameter as in the UV regulator. Following this correspondence, we found the exact matching between the entanglement entropy calculated by the RT formula and the one from the Rindler method in general dimensions. Also we got an explicit relation between the UV cutoff $\delta$ and the geometric cutoff $\epsilon$. This correspondence gives a thorough interpretation for the inconsistency between the universal terms from the Rindler method and the RT formula.

We fail in exploring the relation between the UV and geometric cutoffs for annuli. It is not clear whether the modular Hamiltonian for the annulus is local or not. When the ratio $R_2/R_1$ is above the critical value, the PEE and the MI given by the same linear combination deviates with each other. This difference may be understood from a thorough study about the relation between the UV and geometric regulators. Like the RT surface of the annulus, this relation between the two types of regulators may also undergo a phase transition at the critical point. We hope to come back to this point in future publications.

## Acknowledgments

The authors acknowledge Hongguang Liu, Tatsuma Nishioka, Huajia Wang, Zhuo-yu Xian and Gang Yang for helpful discussions. Especially we would like to thank Pablo Bueno for a careful reading of the manuscript and many useful comments. QW thank the Graduate School of China Academy of Engineering Physics very much for hospitality during the early stage of this work. QW would also like to thank Chang-pu Sun and Chuan-jie Zhu for support. QW is supported by the "Zhishan" Scholars Programs of Southeast University. MH receives support from the US National Science Foundation through grant PHY-1602867 and PHY-1912278, and a start-up grant at Florida Atlantic University, USA.

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
