# Peer review of "Entanglement entropy from entanglement contour: higher dimensions"

_SciPost Physics, doi:SciPost Phys. Core 5, 020 (2022)_

## Round 1 · Referee Report · Anonymous (Referee 1) · 2022-1-26

Strengths

1 The work explore an understudied topic
2 It is clear and emphasize very well which is the physical problem

Weaknesses

1 It is not very well written in the sense that has a lot of typos

Report

In this manuscript the authors explore the differences between the two commonly used methods to regularize the infinite value of the entanglement entropy in quantum field theory in $d\geq 3$ dimensions. The way to study this issue is by analyzing the entanglement contour and partial entanglement entropy from a formal point of view. Also, two examples were done, the case of spherical regions and annuli, in the former an explicit relation between both regulators was computed while in the second example there was not a clear relation between them.

I think the topic is interesting and understudied in the literature and I recommend it for publication.

Requested changes

  1. Along the whole work there are a lot of typos and sentences that are not well written. I appreciate if authors can fix them.

  2. After equation (7) it is mentioned the ALC proposal. I think here the authors should mention that ALC means additive linear combination because the proposal is explained a page later.

  3. Equation (10) as is written seems a derivation of the requirement 7 but is not. What authors are saying is that ${\cal{I}}(A,B)={\cal{I}}(B,A)$ implies that $s_{A\cup C}(A)=s_{B\cup C}(B)$ right?

  4. In the first paragraph of subsection 3.1 there is a reference to a figure that do not show up in the pdf.

  5. In subsection 3.2, in the third point at the beginning of page 11 the authors mention an issue with the ALC proposal that I don't understand. I think what happens in equation (12) is the same that happens with mutual information, the left hand side es UV finite and each term on the right hand side is UV divergent but the combination of them is finite. I don't see which is the issue. Can the authors clarify why this is a disadvantage of the geometric regularization?

  6. In equation (41) the surface ${\cal{E}}_i$ is introduced. From the sentence after the equation is not clear to me how it is defined in terms of ${\cal{E}}$ and the modular slices. Can the authors give a more precise definition of this quantity?

  7. At last a general question out of curiosity. In the examples the computations were done using the ALC proposal, for which a high amount of symmetry is needed (this is why for the spherical regions both regulators were compared successfully). This means that the entanglement contours were obtained by knowing the entanglement entropy of the subregions. Do the authors know if some results along the lines of this work can be obtained just by direct computation of entanglement contours?

  • validity: ok
  • significance: ok
  • originality: good
  • clarity: good
  • formatting: perfect
  • grammar: acceptable

Author:  Qiang Wen  on 2022-02-21  [id 2226]

(in reply to Report 1 on 2022-01-26)

Reply to referee 1: We would like to thank the referee very much for the recommendation for publication. In the following we reply to all the requested changes from the referee:

  1. Along the whole work there are a lot of typos and sentences that are not well written. I appreciate if authors can fix them. Reply: We try to fixed all the typos we can find and rewritten some of the not-well-written sentences.

  2. After equation (7) it is mentioned the ALC proposal. I think here the authors should mention that ALC means additive linear combination because the proposal is explained a page later. Reply: We fixed this in the new version.

  3. Equation (10) as is written seems a derivation of the requirement 7 but is not. What authors are saying is that $\mathcal{I}(A,B)= \mathcal{I}(A,B)$ implies that $s_{A\cup C}(A)=s_{B\cup C}(B)$ right? Reply: Yes, we have rewritten this equation to prevent further misunderstanding.

  4. In the first paragraph of subsection 3.1 there is a reference to a figure that do not show up in the pdf. Reply: Thanks for pointing that out. The reference is now deleted.

  5. In subsection 3.2, in the third point at the beginning of page 11 the authors mention an issue with the ALC proposal that I don't understand. I think what happens in equation (12) is the same that happens with mutual information, the left hand side is UV finite and each term on the right hand side is UV divergent but the combination of them is finite. I don't see which is the issue. Can the authors clarify why this is a disadvantage of the geometric regularization? Reply: Although the divergent part of the entanglement entropies will cancel with each other either we use the UV regulator or different geometric regulators, the remaining finite terms will indeed depend on the regularization schemes. The validity of the ALC proposal needs a suitable choice for the regulators. If we only consider the UV cutoff, usually the finite terms are universal and independent from the UV cutoff, hence the remaining finite terms in the linear combination are insensitive to the cutoff. However, here we also need to consider the geometric regulators and the ``suitable’’ regulator that guarantee the validity of the ALC proposal is indeed a geometric one. We revised the third point to clarify this.

  6. In equation (41) the surface $E_i$ is introduced. From the sentence after the equation is not clear to me how it is defined in terms of $E$ and the modular slices. Can the authors give a more precise definition of this quantity? Reply: The statement is that, for any point $P$ in $A$, its contribution to $S_{A}$ is represented by its partner point $\tilde{P}$ on $E_{A}$. The pair of points $P$ and $\tilde{P}$ are connected by a geodesic normal to $E_{A}$. This one-to-one correspondence between the points on $A$ and $E_{A}$ perfectly matches our interpretation of the entanglement contour. In the same sense, the points in any subinterval $A_i$ in $A$ has partner points forming a geodesic chord $E_{i}$ on $E_{A}$. We rewritten several sentences above Eq.(41) to make this more clear. And I also added references where more details of this construction can be find.

  7. At last a general question out of curiosity. In the examples the computations were done using the ALC proposal, for which a high amount of symmetry is needed (this is why for the spherical regions both regulators were compared successfully). This means that the entanglement contours were obtained by knowing the entanglement entropy of the subregions. Do the authors know if some results along the lines of this work can be obtained just by direct computation of entanglement contours? Reply: We admit that the ALC proposal relies on subset entanglement entropies. The other way to calculate PEE without subset EEs is the EMI formula (15) derived by Casini and Herta in [38], and explicit calculations using the EMI has been done in the vacuum CFT in general dimensions by Casini etc. in a series of papers (for example [40-44]). Nevertheless they usually interpret the EMI as the mutual information in some assumed CFTs where the MI is additive, which does not exist except the 2d free fermions. In cases where both of the ALC proposal and the EMI formula apply, they give the same PEE (see [27] for an explicit calculation and checking). See page 8 for more comments on the EMI formula. Another case is how we derive the contour for spherical regions in section 4, where we used the Rindler method and the geometric construction.

---

## Round 1 · Referee Report · Anonymous (Referee 2) · 2022-2-16

Strengths

1- the paper provides an extended discussion of an interesting topic, clarifying some issues about the partial entanglement entropy.

Weaknesses

1 - remarkable new analytic results do not occur

Report

The authors mainly investigate the partial entanglement entropy in relation with the contour of the entanglement entropy,
which are closely related quantities introduced to study the spatial distribution of the entanglement within the subregion.
The main result concerns the so called “additive linear combination” proposal, which is explored in more detail and better clarified with respect to the previous papers one this topic, although the simplifying assumption of “quasi one-dimensional configurations” is employed in higher dimensions.

I think that the paper contains interesting discussions and deserves the publication. However, my support is conditioned by the following two major changes indicated by 1) and 2) in the dedicated box.

Requested changes

1) About eqs. (50) and (53), the authors should clearly state the regime of validity of these formulas, otherwise the reader could think that they provides the entanglement entropy and the contour for a generic annulus in any model. I think that this is not the case because one expects that this result is model dependent (see e.g. the holographic calculation), while eqs. (50) and (53) have only a geometric nature. Also the comparison between these results and the corresponding holographic calculation in sec. 6.2 should be clarified better.

2) In the introduction, the reader has the feeling that the mutual information is just a geometric way to regularise the entanglement entropy. This is not the case because, for instance, the mutual information contains information that are not encoded in the entanglement entropy. This has been clarified e.g. in 1+1 CFT (see e.g. 0905.2069 and 1011.5482) and should be somehow remarked.

Other suggestions to improve the manuscript are the following

Some other issues/missing references are suggested below:

A) the constant $c$ in Wald formula (35) should be clearly defined and discussed, otherwise it can be confused with the central charge in two dimensional CFTs.

B) In various parts of the paper the connection between the entanglement entropy contour and the holographic bit threads in the context of AdS/CFT has been invoked. As far as I know, this connection has been first proposed during the It-From-Qubit workshop in Bariloche (https://www.youtube.com/watch?v=nO0gL4TZfy8), one year earlier than ref. [35].

C) At the beginning of sec. 6.2, the refs [74], [75] (also the missing refhep-th/0608213 should be mentioned for the HEE viewpoint) Should be also mentioned, where the holographic calculation of the area of the minimal surface in AdS(4) attached to the annulus has been done.

Some minor typos are:

a) Pag. 6, beginning of the paragraph below eq. (8): a typo in the word “seprated”

b) Pag. 7, beginning of the paragraph above eq. (12): the word “proposal” in italics occurs twice.

c) Pag. 9, third line of sec. 3.1: missing reference to the figure

d) Pag. 11, paragraph beginning with “In 2-dimensional theories…” a space is missing before the word “these”

e) Pag. 26, eq. (78) , first term in thee r.h.s.: in the numerator useless round parenthesis occur

  • validity: ok
  • significance: ok
  • originality: low
  • clarity: good
  • formatting: good
  • grammar: good

Author:  Qiang Wen  on 2022-02-21  [id 2227]

(in reply to Report 2 on 2022-02-16)

Reply to referee 2: We thank the referee very much for useful comments and suggestion. In the following, we reply the requested changes from the referee. After that we give an apology for the originality of this paper. We hope the referee can appreciate it.

1) About eqs. (50) and (53), the authors should clearly state the regime of validity of these formulas, otherwise the reader could think that they provides the entanglement entropy and the contour for a generic annulus in any model. I think that this is not the case because one expects that this result is model dependent (see e.g. the holographic calculation), while eqs. (50) and (53) have only a geometric nature. Also the comparison between these results and the corresponding holographic calculation in sec. 6.2 should be clarified better.

Reply: Thanks for pointing this out. The results (50) and (53) are special for the vacuum state of a CFT on a hyperplane. Indeed these results can be reproduced by the EMI formula (15), which was checked by [27] (published in 2020, but appear on arxiv later than this manuscript). We added some clarification below (53), at the beginning of section 5 and also in the abstract. It is hard to say that, they depend on more details of the theory, since they were derived based on the contour function for spherical regions of the vacuum state CFT on a plane with no further input from the theory. It does not make too much sense to compare these results with the holographic one, as they are regulated by different kinds of regulators, hence excluding different types of correlations. Unless an explicit relation between the two types of regulators can be discovered as in the spherical cases. As we pointed out that in the case of annulus or even the case of strips, no such relation shows up using simple geometric tools like the normal geodesics. This maybe consistent with the fact that, the Rindler method does not extend to even the simple regions like strips. The main difference between the two results is that, unlike the holographic result, the entanglement entropy for annulus evaluated from PEE does not undergo a phase transition. We discussed this point in the last three paragraphs in the conclusion section. We suspect that the phase transition may take place in the relation between the UV and geometric regulators, which was not well-understood in this paper. We hope to come back to this point in the future.

2) In the introduction, the reader has the feeling that the mutual information is just a geometric way to regularise the entanglement entropy. This is not the case because, for instance, the mutual information contains information that are not encoded in the entanglement entropy. This has been clarified e.g. in 1+1 CFT (see e.g. 0905.2069 and 1011.5482) and should be somehow remarked. Reply: Thanks, the referee is right. We added a footnote in page 4 to clarify this.

Some other issues/missing references are suggested below: A) the constant c in Wald formula (35) should be clearly defined and discussed, otherwise it can be confused with the central charge in two dimensional CFTs. Reply: We added a sentence to clarify this point below (35).

B) In various parts of the paper the connection between the entanglement entropy contour and the holographic bit threads in the context of AdS/CFT has been invoked. As far as I know, this connection has been first proposed during the It-From-Qubit workshop in Bariloche (https://www.youtube.com/watch?v=nO0gL4TZfy8), one year earlier than ref. [35]. Reply: We have checked this video by Prof. Erik Tonni. The referee is right. We added a footnote in page 15 to stress this point.

C) At the beginning of sec. 6.2, the refs [74], [75] (also the missing ref hep-th/0608213 should be mentioned for the HEE viewpoint) Should be also mentioned, where the holographic calculation of the area of the minimal surface in AdS(4) attached to the annulus has been done. Reply: The referee’s proposal is more reasonable, we have added the references.

Some minor typos are … Reply: Thanks for finding these typos. They were fixed in the new version.

Apology for the originality: The referee mainly mentioned our generalization of the ALC proposal to higher dimensions for the quasi-one-dimensional configurations. But this is only one target of this paper. The main point of this paper is the classification of the geometric regulators in the context of PEE under a (relatively) formal point of view.
1, We pointed out that the geometric regulators exclude different types of correlations from the UV regulator. 2, We evaluate entanglement entropy using PEE with geometric regulators. More importantly, we pointed out that, the well-known Rindler method and the EMI formula should be understand in the context of the PEE with geometric regulators. 3, We clarify the subtleties when comparing the holographic results with those from the Rindler method or the EMI formula. These subtleties appear in many important papers, like [22][57] and other papers (like [38][40][43]) evaluating the EE using EMI. 4, An explicit matching between the EE evaluated using the geometric and UV regulators was done for spherical regions in the vacuum.

We think the discussions above are new to the community and shed some light on the understanding of some old and fundamental subtleties from a new perspective. We hope these will convince the referee to support our publication on Scipost Physics.

---

## Round 2 · Referee Report · Anonymous (Referee 1) · 2022-3-1

Report

I think now the paper is ok for publication but I think this work does not meet the criteria of SciPost and I think can be better placed on JSTAT or PRD.

---

## Round 2 · Referee Report · Anonymous (Referee 2) · 2022-3-15

Strengths

the same of my previous report

Weaknesses

the same of my previous report

Report

The authors have implemented the suggested changes in a satisfactory way. However, I think that the paper is more suitable for Scipost Core rather than Scipost Physics.

---

## Round 2 · Author Response

We thank the referees for suggestions that help improve the manuscript in various aspects. We have made all changes required by the referees, fixed the typos and improved the presentation.

---

## Round 2 · List of Changes

We made the changes required by the referees, including 1, Eq.(10) rewritten 2, subsection 3.1, reference for the Fig deleted. 3, the third point at the beginning of page 11 rewritten 4, paragraph around (41) rewritten 5, clarification added for the validity of (50) and (53) 6, footnote added in page 4 7, footnote added in page 15 8. reference added at the beginning of sec. 6.2 see our reply to the referees for more details.

We changed the title of section 6, added some references and did some minor changes to improve the repsentation.

---

## Editorial Decision

published